# Structure of human spermine oxidase in complex with a highly selective allosteric inhibitor

Elsie Diaz[1,8], Suraj Adhikary[1,8], Armand W. J. W. Tepper[2,8], Daniel Riley[1], Rodrigo Ortiz-Meoz[1], Daniel Krosky[1], Christophe Buyck[3], Carolina Martinez Lamenca[3], Josep Llaveria [4], Lichao Fang [5], Jay H. Kalin [1], Vincent N. A. Klaren[2], Shorouk Fahmy[2], Paul L. Shaffer[1], Robert Kirkpatrick [1], Rodrigo J. Carbajo [4], Maren Thomsen[6] & Antonietta Impagliazzo [2,7✉]

Human spermine oxidase (hSMOX) plays a central role in polyamine catabolism. Due to its association with several pathological processes, including inflammation and cancer, hSMOX has garnered interest as a possible therapeutic target. Therefore, determination of the structure of hSMOX is an important step to enable drug discovery and validate hSMOX as a drug target. Using insights from hydrogen/deuterium exchange mass spectrometry (HDX-MS), we engineered a hSMOX construct to obtain the first crystal structure of hSMOX bound to the known polyamine oxidase inhibitor MDL72527 at 2.4 Å resolution. While the overall fold of hSMOX is similar to its homolog, murine N1-acetylpolyamine oxidase (mPAOX), the two structures contain significant differences, notably in their substrate-binding domains and active site pockets. Subsequently, we employed a sensitive biochemical assay to conduct a high-throughput screen that identified a potent and selective hSMOX inhibitor, JNJ-1289. The co-crystal structure of hSMOX with JNJ-1289 was determined at 2.1 Å resolution, revealing that JNJ-1289 binds to an allosteric site, providing JNJ-1289 with a high degree of selectivity towards hSMOX. These results provide crucial insights into understanding the substrate specificity and enzymatic mechanism of hSMOX, and for the design of highly selective inhibitors.

[1] Janssen Research & Development, Welsh & McKean Roads, Spring House, PA 19477-0776, USA. [2] Janssen Vaccine and Prevention, Archimedesweg 4-6, 2301 CA Leiden, The Netherlands. [3] Janssen Research & Development, Turnhoutseweg 30, B-2340 Beerse, Belgium. [4] Janssen Research & Development, Janssen-Cilag, Discovery Chemistry S.A. Río Jarama, 75A, 45007 Toledo, Spain. [5] Janssen Research & Development Center Medicinal Chemistry, Shanghai 201210, China. [6] Proteros biostructures GmbH Bunsenstr 7a, D-82152 Martinsried, Germany. [7] Present address: Genmab B.V. Uppsalalaan 15, 3584 CT Utrecht, The Netherlands. [8] These authors contributed equally: Elsie Diaz, Suraj Adhikary, Armand W. J. W. Tepper. ✉email: anim@genmab.com

Polyamines (PAs) are ubiquitous in all living cells, and their positive charges allow them to bind DNA, RNA, proteins, and acidic phospholipids. Through these interactions, PAs modulate numerous cellular functions including cell growth, proliferation, differentiation, migration, gene regulation, the synthesis of proteins and nucleic acids, and maintain general cellular oxidative homeostasis[1,2]. The concentrations and chemical natures of the various PAs are crucial for their optimal function[3,4], and as a result, the synthesis, catabolism, and transport of PAs are tightly regulated[1].

There are two distinct but interconnected PA catabolic pathways (Fig. 1), both of which contain oxidases that generate reactive oxygen species (ROS) in the form of $H_2O_2$. The first pathway is a two-step process where both spermidine (Spd) and spermine (Spm) are acetylated at their N1 positions by spermidine/spermine N1-acetyltransferase (SSAT)[5] to form N1-acetylated spermidine or spermine (N-AcSpd and N-AcSpm). These acetylated PAs are either excreted from the cell through their specific transport systems or oxidized by N1-acetylpolyamine oxidase (PAOX) in the peroxisome, resulting in $H_2O_2$, 3-acetoamidopropanal (3Ac-AP), and either putrescine (Put) or Spd. The second pathway is a one-step reaction where Spm is directly oxidized to Spd via the flavin-dependent enzyme spermine oxidase (SMOX; previously PAOh1)[6,7], localized in the cytoplasm and the nucleus[8], to produce Spd, $H_2O_2$, and 3-aminopropanal (3-AP)[9].

In recent years, studies have linked the polyamine metabolic pathway to cancer disease and progression[10,11]. Elevated PA levels are associated with several epithelial cancers through their contribution to cell proliferation, in normal and in neoplastic tissue[2,10,12]. Furthermore, PAs also exert an immunosuppressive effect that can contribute to tumor evasion of the immune response[13–15].

For several epithelial cancers, strong correlations have been established between chronic inflammation, cancer initiation, and progression, and increased levels of hSMOX, which lead to amplified $H_2O_2$ production, oxidative stress, and DNA damage[8,16–20]. In addition, recent work has shown that hSMOX promotes Helicobacter pylori induced carcinogenesis by causing inflammation, DNA damage, and activation of β-catenin signaling[21].

Increased hSMOX levels can also lead to increased concentrations of 3-AP, which can spontaneously decompose into the Michael acceptor acrolein, which can contribute to carcinogenesis through several mechanisms[3].

It has been shown that hSMOX is highly induced by a variety of stimuli, including the inflammatory cytokines tumor necrosis factor-α, interleukin-1β, and interleukin-6, and that the majority of the associated detrimental cellular ROS production triggered by these proinflammatory signals is caused by the subsequent increase in hSMOX-mediated PA catabolism[17,20,22,23].

The robust connection between hSMOX and cancer has led to an interest in targeting its function for therapeutic effect. To date, reported hSMOX inhibitors comprise MDL72527, an irreversible polyamine oxidase inhibitor[24–27], 3,5-diamino-1,2,4-triazole analogs and N1-nonyl-1,4-diaminobutane[28,29]. hSMOX inhibition by these compounds leads to reduced Spd levels, decreased inflammation and DNA damage, a slowing of tumor proliferation, and reduced tumor numbers in several in vivo models[30]. It has also been reported that MDL72527 reduces neuronal death and retinal degeneration, suggesting that blocking hSMOX could effectively prevent or delay vision loss in diabetic patients[22,31,32].

However, while MDL72527 has proven very useful in basic research, it is generally limited as both a therapeutic and a tool to validate hSMOX as a potential drug target due to its high promiscuity, low potency, and cytotoxicity[24,33,34]. The recently published inhibitors 3,5-diamino-1,2,4-triazole analogs and N1-nonyl-1,4-diaminobutane are not hSMOX selective and inhibit the flavin-dependent amine oxidase lysine-specific demethylase 1 (LSD1, also known as KDM1A) and hPAOX, respectively. Thus, to validate hSMOX as a therapeutic target and provide starting points for hSMOX drug discovery, potent and selective hSMOX inhibitors and structural studies to aid in advanced drug design are needed. While human LSD1 has been previously crystallized, only the murine, Zea mays PAOX, and Saccharomyces cerevisiae yeast PAOX structures have been solved[35–38]. Despite several crystallization attempts, only an hSMOX structure model has been reported to date[39,40].

Herein, we describe the structures of hSMOX in complex with MDL72527 and a novel selective allosteric inhibitor (JNJ-1289)

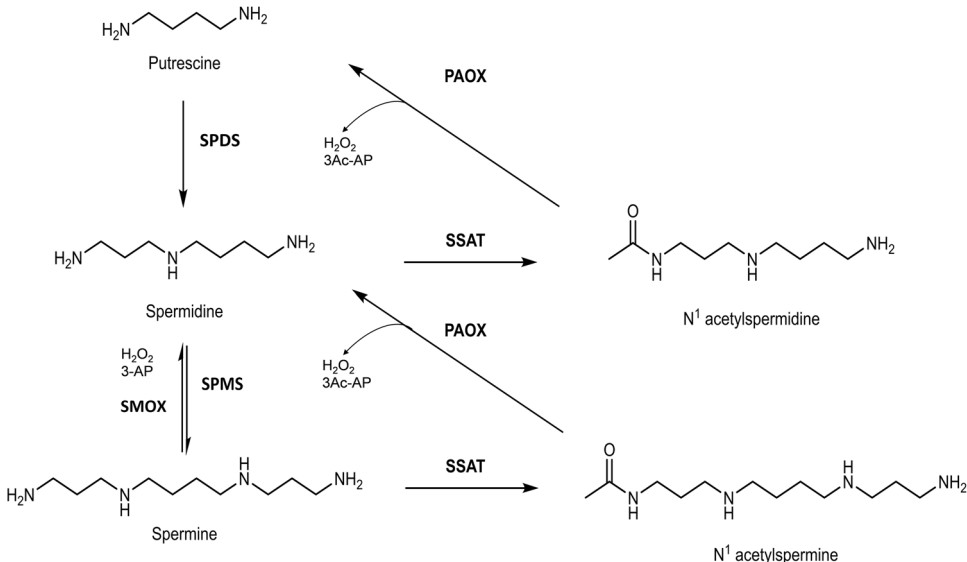

**Fig. 1 Polyamine catabolic pathways.** The two intersecting PA catabolic pathways are depicted. The first pathway involves SSAT that produces N1-acetylated PAs, which are either excreted from the cell through their specific transport systems or oxidized by PAOX, resulting in Put or Spd, $H_2O_2$ and 3Ac-AP. The second pathway involves a one-step reaction where Spm is directly oxidized by hSMOX to produce Spd, $H_2O_2$ and 3-AP. SPDS (spermidine synthase) and SPMS (spermine synthase) are the two enzymes that, through the addition of the aminopropyl group to putrescine and spermidine, generate spermidine and spermine, respectively.

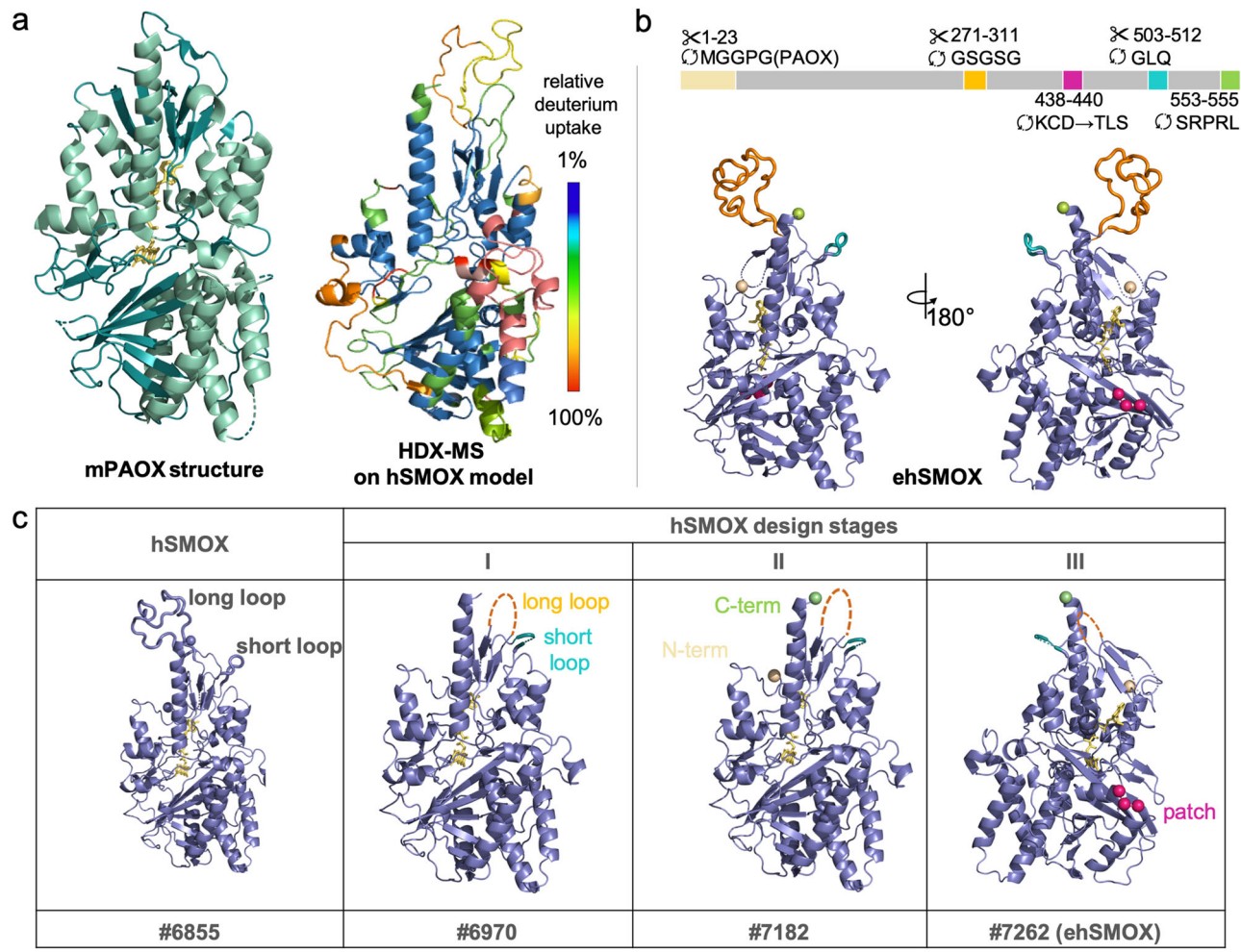

**Fig. 2 hSMOX crystallography construct design. a** Ribbon structure representation in green of mPAOX (PDB ID:5LEA) on the left and dynamic properties of hSMOX analyzed by HDX-MS on the right. The relative deuterium uptake illustrating the local dynamics of different structural units is depicted on hSMOX model using a color code from blue (1% deuterium uptake, protected) to red (100% deuterium uptake; disordered/flexible). **b** Front and back of engineered hSMOX with key modifications depicted and color-coded. **c** Construct modifications for each design stage depicted on ribbon hSMOX model structure and color-coded.

identified by high-throughput screening. This study reveals the structural features of hSMOX that determine its activity and highlights the differences between hPAOX and LSD1, demonstrating that the development of highly selective inhibitors for hSMOX is feasible.

## Results

**Protein design and characterization.** Wild-type (wt) hSMOX has been the subject of many crystallization attempts, and further attempts in our lab were unsuccessful. To generate hSMOX constructs that would be better suited for structural studies, we used an hSMOX model[39,40] derived from the structure of the homologous enzyme murine PAOX (mPAOX; PDB ID:5LAE) to identify structural features that may be hampering crystallization (Fig. 2a). This hSMOX model contains longer and potentially disordered loops absent in mPAOX, and we hypothesized that these loops could interfere with optimal crystal growth. Using the rational design, we replaced these loops in the hSMOX sequence (#6855 hSMOX; Q9NWM0-1 UniProtKB) with the equivalent regions in mPAOX. A general criterion used to minimize any undesired effects of the engineering process around the substrate-binding pocket or on the overall structure of the protein was to introduce mutations in protein segments deprived of well-defined

structure or interactions with other protein segments (Fig. 2b, c and Supplementary Fig. 1).

In the first design phase (Fig. 2c, Stage I) we aimed to shorten the long and short loops (aa 271–311 and aa 503–512, respectively) in the top region of the protein. The long loop was replaced with a linker (GSGSG), and the short loop was replaced with a sequence derived from mPAOX, resulting in construct #6970. Under several crystallization conditions, construct #6970 did not yield good X-Ray quality crystals. To further improve our construct design, we then turned to HDX-MS to experimentally assess the overall conformational dynamics of hSMOX. HDX-MS can be used to glean information on exchange rates for backbone amide hydrogens in solution under physiological conditions[41–43]. Regions that are more flexible or prone to "breathing motions" will display faster and more extensive hydrogen/deuterium (H/D) exchange than stably folded or protected regions. Relative H/D exchange rates mapped onto the hSMOX model are shown in Fig. 2a (Supplementary Fig. 2). These experiments confirmed the flexibility of the two loops modified in Stage I, and identified the N- and C-terminus regions as unstructured elements. Therefore, in Stage II, we replaced the first 23 amino acids at the N-terminus and the last 3 amino acids at the C-terminus with the equivalent sequence in the mPAOX structure (PDB ID:5LAE), resulting in construct #7182, which

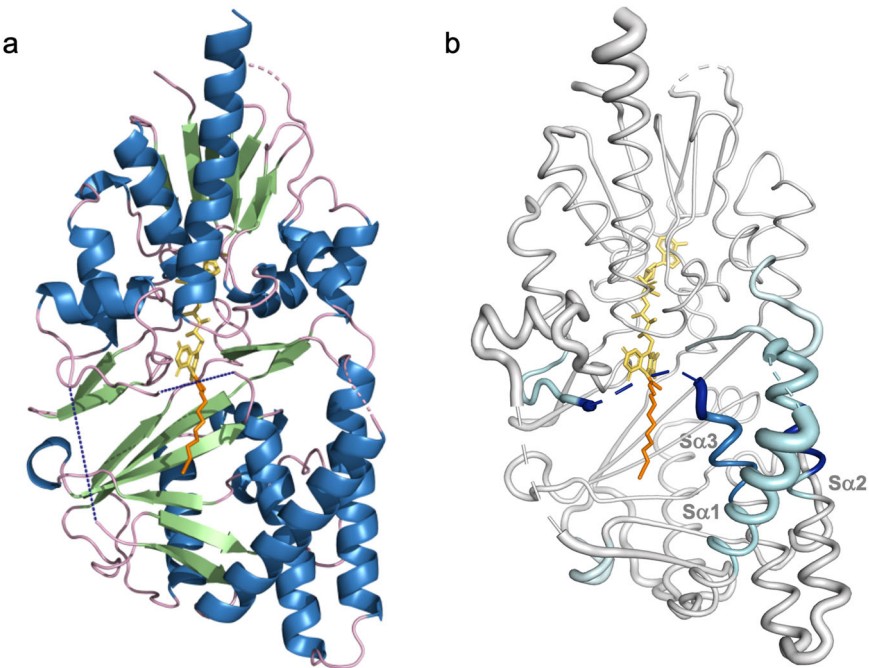

**Fig. 3 Crystal structure of ehSMOX and HDX-MS. a** Structure of ehSMOX with FAD in yellow sticks and MDL72527 in orange sticks. The blue dashed lines represent undefined residues 81–97 (on the left) and 190–210 (in the center). **b** HDX-MS signature differences between apo hSMOX and hSMOX bound to MDL75257 overlaid onto a putty representation (areas with increased diameter correlate with higher B-factors; the dotted blue line represents residues 190–210) of hSMOX; decreased deuterium uptake upon MDL72527 binding, is depicted with a color spectrum from cyan to dark blue with increased protection.

again remained refractory to crystallization (Fig. 2c). In Stage III, we aimed at identifying regions with large and surface-exposed charged residues to apply the principle of surface entropy reduction (SER)[44,45] to favor crystal formation by promoting interactions between oligomers. One of the candidate patches formed by residues Lys438-Cys439-Asp440 was replaced with smaller and less polar residues present at the same position in the mPAOX sequence, respectively Thr382-Leu383-Ser384, to generate construct #7262 (Fig. 2b, c and Supplementary Fig. 1). This last construct will be referred to in the rest of the manuscript as engineered hSMOX: ehSMOX.

Engineered constructs have specific activities within twofold that of wt hSMOX (Supplementary Fig. 3). Thus, the active site structure and other regions critical for catalysis in ehSMOX adequately reflect those of the wild-type enzyme.

**Structure of ehSMOX**. Through standard sparse matrix screening, we identified conditions that supported the growth of single ehSMOX crystals suitable for data collection. These crystals contain one monomer in the asymmetric unit and diffract to ~2.4 Å resolution (Fig. 3a; statistics of data collection and refinement are listed in Table 1). Most of the protein has well-resolved electron density except for two loop regions close to the active-site pocket, presumably due to their flexibility, supported by HDX-MS results.

The overall folding architecture of ehSMOX (Fig. 3a) is similar to other flavoenzymes[46], such as mPAOX[35], and zPAOX[36]. The RMSD between ehSMOX and mPAOX (PDB 5LGB) is 1.7 Å across 395 Cα's, with the most significant differences in the more mobile S-domain. The monomeric enzyme consists of two domains (the FAD containing F-domain and the substrate-binding S-domain) with the non-covalently bound prosthetic FAD group in an elongated conformation at the interface between the domains. As in mPAOX, the FAD isoalloxazine ring is observed in a non-planar twisted conformation leaving the N5

flavin atom pointing towards the substrate-binding site. No obvious liabilities induced by the engineered protein features impact the overall structure: the termini and truncated loops are all exposed to a large solvent channel in the crystal lattice, as is the SER mutant patch.

The FAD-binding domain containing the adenosine monophosphate component of FAD comprises residues 3–44, 67–76, 219–304, and 435–490. The FAD prosthetic group is deeply embedded within the structure, and only the isoalloxazine C5a, N5, and C4a atoms are solvent accessible. Like mPAOX, the FAD-containing tunnel is formed by a central parallel β-sheet (strands Fβ1, Fβ2, Fβ11, and 13) flanked by a β-meander (Fβ8, Fβ9, and Fβ10) and three α-helices (Fα1, Fα4, and Fα8) (Supplementary Fig. 4). Additional small helical motifs (Fα2, Fα5, Fα6, and Fα7) and a β-hairpin (Fβ3, Fβ4) surround the outer shell of this domain. The cofactor binding mode is conserved between ehSMOX and mPAOX, involving H-bonds with Val241, Glu35, Arg43 and Trp60, with the only difference being Thr465, which corresponds to Glu465 in mPAOX.

The substrate-binding domain that interacts with the FAD isoalloxazine ring is formed by two parts of the protein (residues 78–217 and 305–424) folded into a 6-stranded mixed β-sheet (Sβ1–6) flanked by five α-helices (Sα1–5) (Supplementary Fig. 4). The ehSMOX electron-density map corresponding to residues 81–97 and 190–210 (both in dark dashed lines in Fig. 3a) could not be built due to the lack of electron density observed. As shown in Fig. 3a, these loops occur on the left side of the substrate entrance and in the area just above the catalytic site, respectively, suggesting these loops may have a possible role in substrate binding. In the absence of well-defined electron density for these two loops, the putative ehSMOX substrate-binding cleft appears open to solvent.

**The active site**. While the central β-sheet structure seems conformationally conserved between ehSMOX and mPAOX, the Sα1–3 surrounding the catalytic tunnel presents significant

**Table 1 Data collection and refinement statistics.**

|  | ehSMOX + MDL72527 | shSMOX + MDL72527 + JNJ-1289 |
|---|---|---|
| PDB entry | 7OXL | 7OY0 |
| Data collection |  |  |
| X-ray source | SLS PX-II | SLS PX-II |
| Space group | P 3 2 1 | P 3 2 1 |
| Unit cell (Å) | 190.92, 190.92, 43.57 | 193.75, 193.75, 44.33 |
| Anisotropic resolution (Å) | 2.63/2.63/2.37 (2.63–2.40) | 2.98/2.98/2.04 (2.41–2.10) |
| Unique reflections | 28,433 (1423) | 27,031 (1354) |
| Redundancy | 11.8 (10.6) | 13.0 (15.5) |
| Ellipsoidal completeness (%) | 93.7 (45.4) | 93.9 (70.3) |
| Mean $I/\sigma_I$ | 9.0 (1.4) | 18.9 (2.0) |
| $R_{pim}$ (%) | 5.1 (57.8) | 2.0 (36.7) |
| CC 1/2 (%) | 99.70 (57.50) | 99.90 (79.60) |
| Refinement |  |  |
| No. of reflections (working/test) | 27,492/941 | 25,687/1345 |
| Number atoms |  |  |
| Protein | 3469 | 3621 |
| Ligand | 0 | 22 |
| FAD + MDL72527 | 67 | 67 |
| Water | 56 | 89 |
| Other | 10 | 1 |
| $R/R_{free}$ (%) | 18.5/21.4 | 19.5/23.9 |
| Deviation from ideal geometry |  |  |
| Bond lengths (Å) | 0.004 | 0.004 |
| Bond angles (°) | 1.36 | 1.37 |

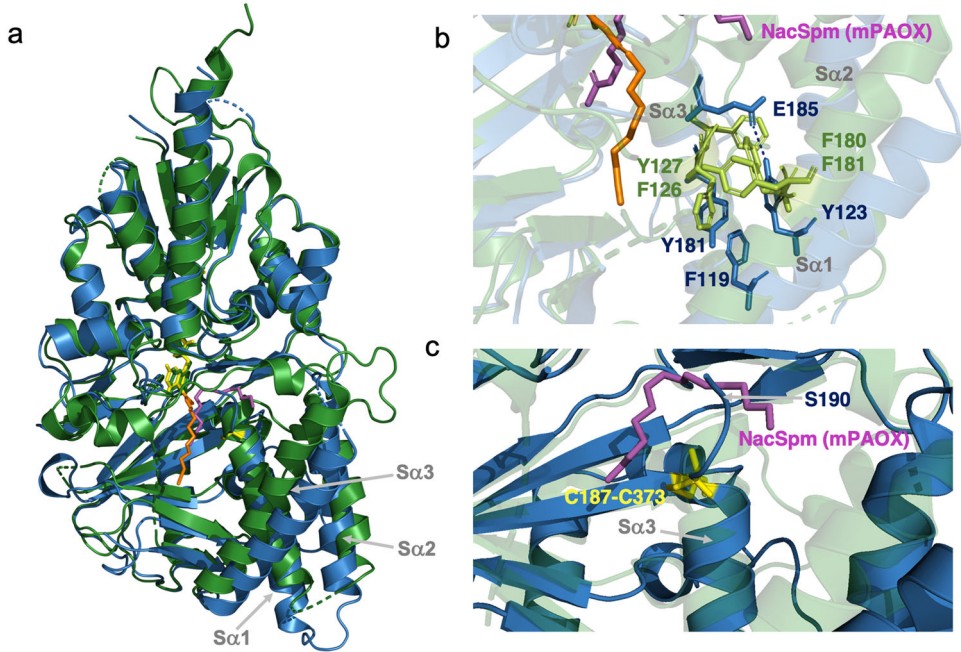

**Fig. 4 Comparison of ehSMOX with mPAOX structures. a** Superimposition of the ribbon representation of ehSMOX (in blue) bound to MDL72527 (in orange sticks) and mPAOX (PDB: 5MBX in green) with its substrate *N*-AcSpm (in magenta sticks). FAD is depicted in yellow and green, respectively for ehSMOX and mPAOX and the ehSMOX cystine bridge (Cys187-Cys373) is shown in yellow sticks. **b** Residues involved in π-interactions are in light green sticks for mPAOX and in blue for heSMOX; the dashed line indicated the additional H-bond between E185 and Y123 only present in ehSMOX. **c** The Sα3 helix of mPAOX (transparent green ribbon) is longer compared to ehSMOX and directed towards the back of the protein. In ehSMOX, the cysteine bridge depicted in yellow sticks indicates the end of helix Sα3 and the beginning of the loop directed in the opposite direction compared to mPAOX. Residue S190 of heSMOX clashes with the N-AcSpm (in magenta stick) superimposed from mPAOX structure.

differences between the two enzymes, both in sequence and structure (Fig. 4 and Supplementary Fig. 5). In the substrate-binding pocket of mPAOX, the Sα1 helix is longer and closer to the main body of the protein compared to ehSMOX (Fig. 4a). In addition, in mPAOX a cluster of aromatic residues involved in π-interactions

(Phe180, Phe181 on Sα1 and Phe126, Tyr127 on Sα3 in Fig. 4b), brings the two helices Sα1–3 close to each other. On the contrary, in ehSMOX, the interaction between the three helices Sα1,2 and 3 seems looser with only one π-interaction between Sα1 and 3 and an additional H-bond between E185 and Y123 (Fig. 4b).

In ehSMOX, a distinctive cystine bridge between Cys187 and Cys373 caps and stabilizes the C-terminal end of the Sα3 helix (Fig. 4c), shortening it relative to the corresponding α-helix in mPAOX, thus possibly positioning the unresolved aa190–210 loop of ehSMOX to protrude more outside the surface of the protein. Indeed Ser189-Ser190 (Sα3) which are the first resolved residue of the aa190–210 loop in ehSMOX, clash with $N$-AcSpm when superimposed with mPAOX (PDB ID:5MBX) (Fig. 4c and Supplementary Fig. 5a), reinforcing the hypothesis that the position of this loop significantly differs between mPAOX and hSMOX. When overlapping the structures of ehSMOX and mPAOX bound to $N$-AcSpm, mPAOX Asn313 which functions as a H-bond donor to the $N1$-acetyl group of $N$-AcSpm, in ehSMOX is occupied by a threonine residue (T309) which would be too far for such a bond to be formed (Supplementary Fig. 5a).

Additional residues that are significant for the difference in the substrate-binding pockets between ehSMOX and mPAOX (in brackets) are Glu188 (Val187); His212 (Asp211), Asp356 (Leu361) and Trp371 (Phe375) (Supplementary Fig. 5b).

Despite the presence of MDL72527 being necessary to form crystals of ehSMOX (but not sufficient to allow crystallization of WT and other engineered constructs), weak electron density of this inhibitor is observed beyond the imine N5 nitrogen of the covalent adduct (Supplementary Fig. 6), similar to the reported MDL72527-mPAOX co-crystal structure[35]. Due to the continuous density from N5 of FAD, we could only attribute this electron density to the formation of a covalent MDL72527/FAD adduct[47], consistent with MDL72527 acting as an irreversible inhibitor. The butterfly shape of the isoalloxazine ring is further evidence that the cofactor is present in a reduced or covalently bound state[48,49]. Part of the apparent flexibility of MDL72527 in this ehSMOX structure can be attributed to the orientation of His62 (His64 in mPAOX) that appears unsuited to form a H-bond with the N5 atom of MDL72527, as occurs in mPAOX (Supplementary Fig. 5). This His62 residue most likely has to adopt several conformations in the hSMOX enzyme, but there is no clear electron density to support the presence of multiple rotamers in the apo ehSMOX structure.

To further understand the absence of clear electron density for MDL72527, we compared the HDX-MS signatures of hSMOX in the presence and absence of MDL72527 (Fig. 3b; overall heat maps are shown in Supplementary Fig. 1). The results revealed an epitope for MDL72527-binding consistent with that seen in the ehSMOX co-crystal structure and reported for mPAOX (PDB ID:5LGB)[35]. While there is decreased deuterium uptake in aa202–239 upon MDL72527 binding, the loop remains relatively dynamic compared to neighboring residues and likely explains the lack of resolution of this loop in the ehSMOX crystal structure. Moreover, only a modest level of protection is observed on ehSMOX Sα1–3 upon reaction with MDL72527, which we interpret as the dynamic nature of MDL72527 within the active site pocket when only tethered by a covalent linkage with FAD.

**High-throughput biochemical screen and hSMOX inhibitor discovery**. To discover potent and chemically tractable inhibitors of hSMOX, we developed a high-throughput assay that detects the enzymatic activity of hSMOX using HyPerBlu (Lumigen), a reagent that measures the concentration of $H_2O_2$, a product of hSMOx enzymatic activity (Supplementary Fig. 7). Through this assay, the apparent $K_m$ value of Spm for hSMOX was determined to be 34 μM. Furthermore, the potency of the hSMOX inhibitor chlorhexidine measured in our HTS assay (IC$_{50}$ = 1.9 μM) is similar to the value reported for the murine SMOX enzyme ($K_i$ = 3.8 ± 0.2 μM; Supplementary Fig. 8)[50]. Based on this assay, we carried out a high-throughput screening campaign searching

for inhibitors of hSMOX against a diverse compound collection. Miniaturized to a 1536-well format, the assay returned Z' scores of 0.72 and signal/background values of 18.4 over multiple screening days. This effort identified 4-((4-imidazo[1,2,α]pyridine-3-ylthiazol-2-yl)amino)phenol (JNJ-1289-C$_{16}$H$_{12}$N$_4$OS 309.08 Da (theoretical value of 308.0731 Da) see Supplementary Data High Resolution Mass Spectoscopy_JNJ-1289) as a highly potent hSMOX inhibitor (IC$_{50}$ = 50 nM under these reaction conditions, Fig. 5a, b). To measure the selectivity of JNJ-1289, we assayed the inhibitor against hPAOX and LSD1 and found that JNJ-1289 has IC$_{50}$ values >2 μM against both enzymes (Fig. 5b). By contrast, chlorhexidine shows robust inhibition of both hPAOx and hSMOX (Supplementary Fig. 9). This observed selectivity against hPAOX and LSD1 differentiates JNJ-1289 from other molecules reported in the literature that inhibit these enzymes[27,28,50,51]. Thermal-shift analysis of hSMOX in the presence of JNJ-1289 or DMSO revealed robust stabilization of the protein ($\Delta T_m$ = 11.3 °C) by JNJ-1289, consistent with JNJ-1289 binding to hSMOX in a specific manner (Fig. 5c).

The inhibition mechanism of JNJ-1289, was further characterized by measuring the apparent potency of the inhibitor without (IC$_{50}$ = 127 nM) and after a 120 min pre-incubation (IC$_{50}$ = 8 nM) of hSMOX and JNJ-1289 prior to substrate addition, suggesting that JNJ-1289 is a time-dependent hSMOX inhibitor (Fig. 5d). Further pre-incubation of JNJ-1289 with hSMOX for 240 min resulted in a potency within twofold of the results observed using a 120 min pre-incubation (Supplementary Fig. 10). We note that this time-dependent behavior is not observed with chlorhexidine inhibition of both hSMOX and hPAOX (Supplementary Fig. 9). We also do not observe time-dependent inhibition of hPAOX by JNJ-1289 (Supplementary Fig. 9). Progress curve analysis was employed to further characterize the inhibition of hSMOX by JNJ-1289. First, carrying out progress curve analysis at varying concentrations of Spm at a fixed inhibitor concentration demonstrated that the apparent first-order binding rate constant ($k_{obs}$) of JNJ-1289 decreased with increasing substrate concentration, suggesting that JNJ-1289 is a competitive hSMOX inhibitor with respect to Spm (Supplementary Fig. 11). Second, measuring the $k_{obs}$ of JNJ-1289 binding to hSMOX using varying concentrations of inhibitor at a fixed Spm concentration revealed that the compound binds to hSMOX via a two-step binding mechanism with an apparent second-order on-rate constant ($k_{on}$) value of $2.5 \times 10^3$ M$^{-1}$ s$^{-1}$, and a calculated first-order dissociation rate constant value ($k_{off}$) of $2.5 \times 10^{-5}$ s$^{-1}$ (Supplementary Fig. 11d). Together, these data suggest that JNJ-1289 initially forms a weak complex with hSMOX with an apparent $K_i$ value of 1.4 μM, followed by a relatively slow protein isomerization that forms the final tightly bound inhibitor-enzyme complex.

To determine whether JNJ-1289 would be able to target hSMOX in a cellular environment, we developed a Cellular Thermal Shift Assay (CETSA®)[52] using human lung carcinoma cells (A549) which endogenously produce high levels of hSMOX[53]. CETSA evaluates the binding of low molecular weight compounds to target proteins directly in intact cells by taking advantage of the increased resistance to thermal denaturation that ligand binding can confer on proteins. Detection of the relative amounts of remaining soluble protein using AlphaLISA after incubation of A549 cells at the optimal melting temperature of hSMOX for CETSA (54 °C) was performed in the presence or absence of hSMOX ligands. A549 intact cells treated with MDL72527 showed a clear AlphaLISA signal increase upon heating compared to vehicle-treated cells, indicating robust cellular hSMOX engagement by this irreversible inactivator. In contrast, neither JNJ-1289, Benspm nor chlorhexidine substantially increased the AlphaLISA signal compared to the vehicle-

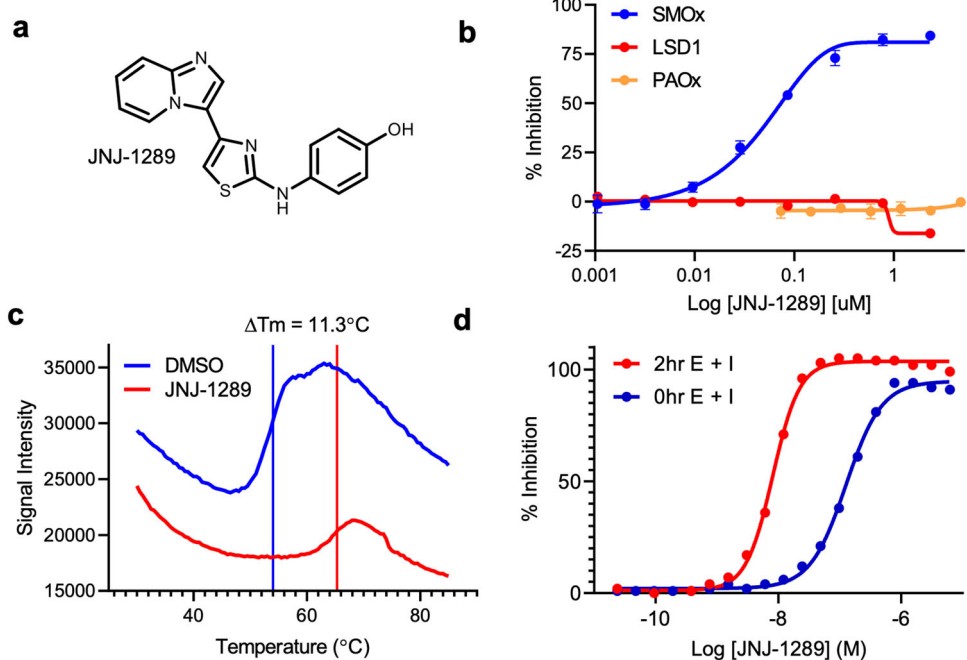

**Fig. 5 JNJ-1289 inhibits hSMOX in a time-dependent manner. a** Structure of JNJ-1289. **b** Inhibition by JNJ-1289 at 2.5 nM hSMOX, 60 pM hPAOX or 10 nM LSD1 in the presence of 30 μM Spm, 20 μM N-AcSpm, or 8 μM H3K4me2 peptide, respectively. JNJ-1289 inhibits hSMOX under these assay conditions with an IC$_{50}$ value of 50 nM. **c** Thermal shift analysis of hSMOX reveals that incubation with JNJ-1289 (red line) stabilizes the protein with $\Delta T_m = 11.3$ °C compared to DMSO control (blue line). **d** Inhibition of hSMOX measured using a 0 h or 2 h preincubation of enzyme and JNJ-1289 at 2.5 nM enzyme and 30 μM Spm. The observed IC$_{50}$ values of 127 and 8 nM at 0 and 2 h enzyme-inhibitor preincubation times, respectively, suggested that JNJ-1289 inhibits hSMOX in a time-dependent manner.

treated controls (Supplementary Fig. 12). These CETSA experiments were repeated at different ligand concentrations and gave similar results, confirming the stabilization of hSMOX in A549 cells by MDL72527 but not JNJ-1289, suggesting an absence of cellular hSMOX engagement by any of these tested noncovalent ligands under the assay conditions.

**Co-crystal structure of JNJ-1289-bound ehSMOX.** To understand the binding of JNJ-1289 to ehSMOX, the inhibitor was soaked into crystals of ehSMOX and diffraction to anisotropic resolution cutoffs of 2.98, 2.98, and 2.04 Å was collected (Table 1). Following molecular replacement using the apo ehSMOX structure, there was clear electron density observed for JNJ-1289 (Supplementary Fig. 13).

The structure of JNJ-1289-bound ehSMOX overlaps well with MDL72527-bound ehSMOX (RMSD 0.61 Å using all main chain atoms). JNJ-1289 binds ehSMOX in an allosteric pocket ~13 Å from the FAD isoalloxazine ring (Fig. 6a). Critically, one of the loops near the active site entrance (aa 190–210), which was not resolved in the first ehSMOX structure, is now well defined and is bent in front of the substrate-binding pocket above JNJ-1289, indicating stabilization of the loop upon inhibitor binding (Fig. 6a).

The conformation of this loop observed in the JNJ-1289-ehSMOX co-crystal structure appears distinct from the analogous loop in the mPAOX structures[35,36]. In both structures, ehSMOX and mPAOX, the loop is tethered to the rest of the protein through several H-bonds stabilizing the two substantially different conformations (Fig. 7a, b).

The loop position in the JNJ-1289 bound-ehSMOX confers a U-shaped substrate-binding cavity that is more similar to the zmPAOX than to the mPAOX structure (Supplementary Fig. 14). The substrate-binding cavity passes through the protein structure at the substrate- and FAD-binding domains interface. This

second structure of heSMOX distinctly shows the end of the Sα3 heSMOX loop protruding in front of the flavin (Supplementary Fig. 5a), forming a bend that changes the tunnel direction towards the protein surface on the right-hand side, creating two cavities unique to ehSMOX (Fig. 7a, b and Supplementary Fig. 15). Interestingly, this structural feature also seems to be present in the MDL72527-bound ehSMOX structure, although the whole loop was unresolved (Supplementary Fig. 5a). The dimensions of the heSMOX tunnel in this co-crystal structure are about 20 Å × 7 Å and the left-hand side of the catalytic site is open due to the undefined loop aa 81–97 (Supplementary Fig. 15), which could adopt a more closed conformation to cap the catalytic site upon substrate binding. The binding pocket for JNJ-1289 observed in the ehSMOX co-crystal structure reflects that of hSMOX in solution obtained using HDX-MS (Fig. 6b and Supplementary Fig. 2). These results confirm that the most protected areas on hSMOX upon JNJ-1289 binding are located below the substrate binding pocket (aa 147–153, 163–173, 208–219, and 484–493) involving the flexible loop aa190–210. The increased protection seen for aa 484–493 suggests that binding of JNJ-1289 stabilizes the interaction between FAD- and substrate-containing domains.

To further analyze the binding of JNJ-1289 in the presence of MDL72527, we carried out a nanoDSF experiments to parse the differences in stabilization offered by the two ligands. The experiments show that both ligands increase the thermal stability of the protein, independently from each other (Supplementary Fig. 16): JNJ-1289 can engage hSMOx independently from MDL72527 in solution and the presence of MDL72527 does NOT affect the stabilization offered by JNJ-1289.

**Description of the inhibitor binding pocket.** The electron density for JNJ-1289 allows unambiguous placement in an allosteric pocket nestled between the protein surface of the 'open'

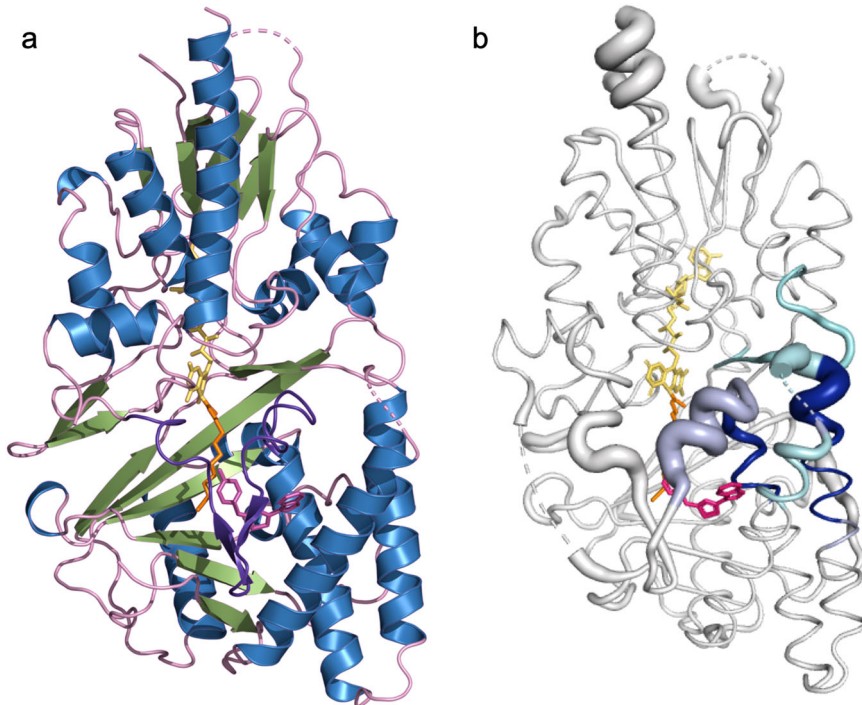

**Fig. 6 Structure of ehSMOX containing the ligand JNJ-1289 and HDX-MS of the complex. a** Structure of ehSMOX in ribbon representation. The FAD cofactor is shown in yellow, MDL72527 in orange and the ligand JNJ-1289 is shown in magenta sticks; the loop in front of the substrate-binding pocket is in purple color. **b** HDX-MS signatures differences between hSMOX and hSMOX bound to JNJ-1289 overlaid onto a putty representation (areas with increased diameter correlate with higher B-factors) of ehSMOX; decreased deuterium uptake upon JNJ-1289 binding, is depicted with a color spectrum from cyan to blue with increased protection.

structure and the previously unresolved aa190–210 loop closing on top. The electron density is weaker for the phenol portion of the inhibitor and this is consistent with it being more solvent exposed and likely to be somewhat mobile in the binding site (Supplementary Fig. 13). The JNJ-1289 phenol oxygen is involved in a H-bond with the carboxylic acid of Glu207 (Fig. 7c). The phenol aromatic ring is flanked by Val197 and partially solvent-exposed, as is the nitrogen between the phenol and the thiazole (Fig. 7c). The central thiazole is involved in several π-stacking interactions with residues Val184 (H-π), Tyr181 (T-shaped π) and Trp205 (π-π). The imidazopyridine ring of the inhibitor forms a H-bond with the carboxylic acid of Glu185, while the pyridine ring makes an edge-to-face interaction with the side chain of Tyr127 (Fig. 7d). Overlaying the mPAOX structure with the ehSMOX co-crystal structure revealed that the shape of the allosteric pocket differs between the two enzymes (Supplementary Fig. 17). In particular, the side-chain of mPAOX Tyr127 would directly clash with JNJ-1289. In addition, the analogous loop to ehSMOX aa190–210, which makes several key interactions with JNJ-1289, may not be able to adopt the same conformation in mPAOX due to differences in the position of Sα1. Hence, the high degree of selectivity of JNJ-1289 for SMOX over PAOX can be rationalized from a comparison of the crystal structures.

One way to optimize inhibitor potency for a target is to match the solution-phase conformation of the compound with its protein-bound conformation. This preorganization of the compound solution-phase conformation reduces the energetic penalties associated with the protein-binding event[54]. To investigate the solution-phase conformation of JNJ-1289, we carried out an NMR study in DMSO as a surrogate for an aqueous polar environment. We found that the NH bond is preferentially oriented towards the S atom, a similar conformation to what is observed in the ehSMOX/JNJ-1289 X-ray pose

(Supplementary Fig. 18), providing rigidity to the central part of the molecule. The imidazopyridine ring shows higher flexibility, adopting several orientations around the bond linking it to the thiazole. Therefore, JNJ-1289 partially adopts the bioactive pose in solution, which to some degree accounts for its high affinity for hSMOX.

**Modeling Spm into ehSMOX/JNJ-1289 co-crystal structure.** We have used this 'closed' crystal form as a starting point to model Spm in the ehSMOX active site. First, we refit the reduced butterfly conformation of FAD to its flatter oxidized form to reflect the form of the enzyme that binds Spm. Second, since Spm is a highly flexible molecule that can exist in many possible conformations in solution, we restrained the model by using knowledge of the catalytic mechanism[35] in which the central nitrogen of Spm binds near the FAD quinone moiety. Third, we made use of Ser463, that is conserved between hSMOX and mPAOX and is involved in a H-bond with N-AcSpm in mPAOX, as a guide to dock Spm[35].

Docking with these restraints followed by manual refinement resulted in a binding pose depicted in Supplementary Fig. 19a, b. Overall, the substrate-binding pocket is negatively charged and complementary to the positively charged Spm substrate, with all Spm nitrogen atoms believed to be protonated in this environment. In our model, all Spm nitrogen atoms are involved in a hydrogen bonding network involving Glu188, Lys460, Tyr461, Tyr462, Ser463. A histidine residue (His64 in mPAOX, His62 in ehSMOX) that is highly conserved in polyamine oxidases has been proposed to play a crucial role in the formation of the hydrogen bonding network that defines the active site[35]. In our ehSMOX model, His62 appears 3.4 Å distant from the hypothetical position of Spm, which is suboptimal for substrate

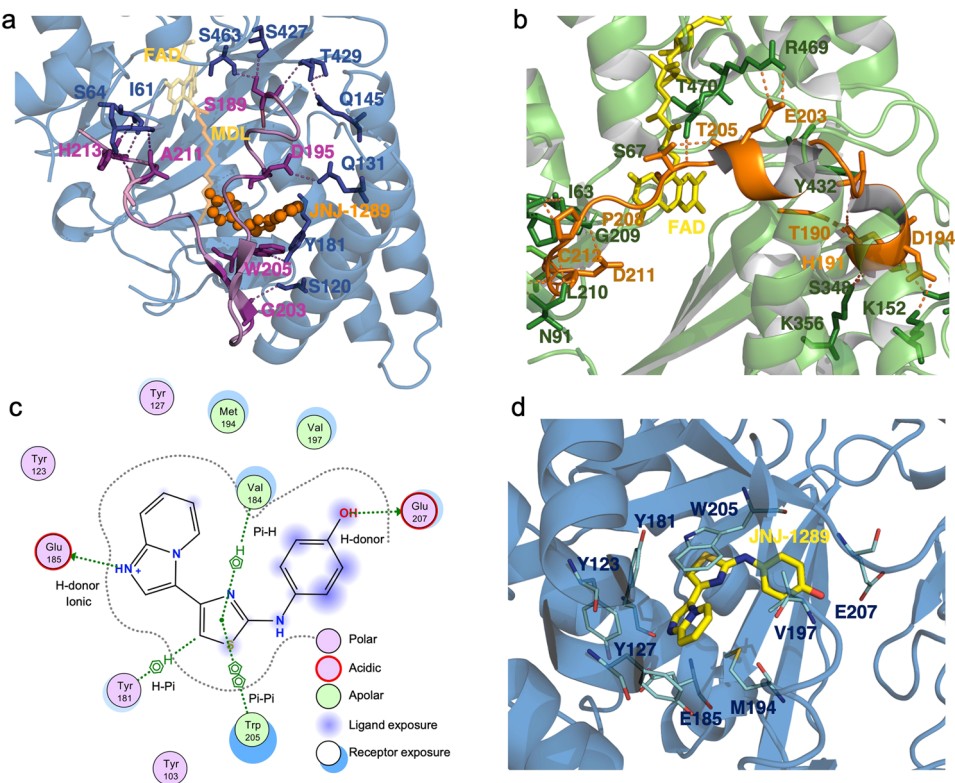

**Fig. 7 ehSMOX and mPAOX substrate loop position and binding pocket of JNJ-1289 on ehSMOX. a** ehSMOX in a blue cartoon with FAD and MDL72527 in yellow and orange transparent sticks; JNJ-1289 in orange spheres. In pink ribbon is depicted the flexible loop above the substrate pocket, in violet sticks, are the residues of that loop involved in intramolecular H-bonds (dotted lines) with residues (blue sticks) in the main body of the protein. **b** mPAOX (PDB: 5MBX) in a green cartoon with FAD in yellow. The loop above the substrate pocket is colored in orange, and orange and in green sticks are indicated residues involved in intramolecular H-bonds (orange dotted lines) belonging to the loop and to the rest of the protein, respectively. **c** JNJ-1289 interactions with ehSMOX residues. **d** JNJ-1289 (in yellow sticks) interactions with ehSMOX represented in blue ribbon. Residues involved in H-bonds (cyan dashed lines) or π-interactions (yellow dashed lines) are depicted in cyan sticks.

interaction (Supplementary Fig. 19c). However, weak electron density suggests that His62 adopts a rotamer similar to that seen in mPAOX with low occupancy in the current structure and could reorient in the presence of substrate.

In an attempt to find similarities and differences between vertebrate and invertebrate polyamine oxidases we used the spermine bound model of heSMOX to compare it with the Spermine bound form of *Saccharomyces cerevisiae* yeast PAOX (yPAOX), *Zea mays* PAOX (zmPAOX) and murine PAOX (Supplementary Fig. 20). While the conserved W60, H62 and K311[35] located near the FAD N5 are occupying a similar position in heSMOX, mPAOX and yeast PAOX, the overall active site appears notably different. Furthermore, spermine in the structures of yPAOX and zmPAOX occupies a different position compared to the spermine bound model of heSMOX, possibly reflecting how both flexibilities of the substrate and of the catalytic site may allow different substrate positions. We emphasize that our spermine binding mode is a model and needs additional experimental confirmation.

Intrigued by the allosteric pocket of JNJ-1289, we have used the MOE Sitefinder functionality to identify alternative druggable sites on ehSMOX structure. The two biggest pockets identified are named SITE1 (yellow) and SITE2 (pink) (Supplementary Fig. 21). SITE1 combines the substrate-binding site where spermine can be modeled in and the allosteric site under the newly formed loop covering JNJ-1289. SITE2 is located in one of the two unique pocket (cavity 1 in Supplementary Fig. 15) at the back of the protein.

## Discussion

Human SMOX contains several flexible and disordered regions, including a long loop absent in murine PAOX, which likely explains the failure of previous crystallization attempts. By explicitly addressing these flexible regions and employing protein engineering strategies to modulate the protein surface properties, we were able to crystalize the protein, determine its structure and to establish similarities and differences with its homolog protein mPAOX. Furthermore, the presence of MDL72527 turned out to be essential to obtain diffracting crystals, likely through a stabilizing effect on the protein structure and improvement in the crystal quality. To further underline the relevance of the present heSMOX crystal structure, we report that our attempts to crystallize the wt hSMOX in complex with several SMOX specific monoclonals antibodies[55], were all unsuccessful.

Despite the similar fold and conservation of key residues, the active site differs remarkably between hSMOX, and other polyamine oxidases (mPAOX, zmPAOX and yPAOX). For example, the position of Asn313 that stabilizes through an H-bond N-AcSpm in mPAOX is occupied by Thr309 in hSMOX which is not able to form a similar bond. Previous work suggested that Glu196 and Ser198 in hSMOX (corresponding to Leu195 and Ala197 in mPAOX) are essential in determining the substrate specificity[39]. In the present structure, these residues are part of a flexible loop at a distance from the N5-FAD longer than 15 Å apart, making it less likely that these residues are critical in determining substrate specificity. Nevertheless, long-range mutations effects on the substrate-binding pocket could explain the discrepancy of such results.

Comparison of the catalytic sites of ehSMOX and mPAOX show differences in the charge and shape of the pockets, which are likely crucial in the different substrate specificities of these enzymes, and delineate the remarkable selectivity of JNJ-1289 for hSMOX over hPAOX. This potent allosteric inhibitor stabilize a uniquely positioned loop in hSMOX, forming several H-bonds that tether the loop to the rest of the protein. The position of this loop, though apparently induced by the ligand, is made possible by the presence of a cysteine bridge on the top of the substrate-binding pocket, a feature that is notably absent in mPAOX. Consequently, the hSMOX active site is present in a unique conformation from PAOX, which would hinder the binding of the N-AcSpm shown in mPAOX structure. Although we have attempted to model the conformation of the spermine substrate in the active site pocket, further studies will be needed to assess the substrate binding mode and the catalytic mechanism experimentally.

Based on our substrate-binding model, Spm was able to dock into the ehSMOX structure at the same time as JNJ-1289. However, our analyses suggest that JNJ-1289 is a competitive inhibitor with respect to Spm. If this substrate-binding model is accurate (i.e., JNJ-1289 and Spm are not orthosteric), apparent mutually exclusive binding could arise through the closure of both the inhibitor and substrate binding sites by the mobile aa190–210 loop induced by the binding of either Spm or JNJ-1289. As observed in the JNJ-1289-ehSMOX co-crystal structure, the compound-induced conformation of aa190–210 closes off the putative active site. Thus, while Spm can be docked into this ehSMOX conformation, it would have a substantial kinetic barrier to diffuse into the now occluded active site. Conversely, Spm binding to hSMOX could antagonize JNJ-1289 binding through induction of a closed conformation of the aa190–210 loop, either blocking direct access to the JNJ-1289 binding site and/or inducing a different loop conformation that is not conducive to inhibitor binding.

JNJ-1289 showed good in vitro cellular permeability (Papp(A-B) = 15 cm/s × $10^{-6}$) in a MDCK-MDR1 (+inh) system, which makes the interpretation of the negative CETSA results confounding, especially since JNJ-1289 induced a substantial increase in the apparent melting temperature of purified hSMOX (Fig. 5c). At this moment, we don't have an explanation for the apparent lack of target engagement of hSMOX by JNJ-1289 in the CETSA assay; however, it appeared relevant to us to report cellular hSMOX engagement by MDL as it has been the irreversible SMOX inhibitor used in many preclinical models published so far[19,29,31].

The data presented here show the systematic, rational engineering of human SMOX to overcome crystallization challenges, to yield the first structures of human SMOX and a first-in-class, selective, allosteric inhibitor of hSMOX. Furthermore, these data work together to explain how hSMOX can be inhibited selectively and enable the development of improved selective hSMOX inhibitors.

## Methods

**Protein design**. PyMOL Molecular Graphics System, Version 1.5.0.3 (Schrödinger, LLC) was used as the molecular visualization software.

**hSMOX expression, purification, and characterization**. All hSMOX constructs were produced in E. coli BL21(D3) transformed with the plasmid PET28b(+) containing the hSMOX isoform 1 gene (SMOX_HUMAN 1-555 UniProtKB - Q9NWM0) or variants thereof. All variants contained an N-terminal 6xhistidine tag to facilitate purification followed by a TEV cleavage site to facilitate tag removal. LB/kanamycin (25 μg/ml) growth medium in flasks shaken at 250 rpm was used throughout. Starting from a preculture, bacteria were grown to reach an $OD_{600}$ value of 0.7, after which protein expression was induced by the addition of IPTG (0.1 mM) at 18 °C overnight. Cells were harvested by centrifugation

(4000 g × 15 min), washed with PBS and resuspended in 10 mM HEPES pH 7.4 + 2x protease inhibitor (Roche cOmplete EDTA free tablets; cat# 372268) and lysed using a OneShot cell disruptor at 2.7 kbar. The lysate was clarified using centrifugation (30 min at 10,400 × g), diluted 10-fold with 10 mM HEPES pH 7.4 + 1x protease inhibitor and loaded onto a 6 ml column containing equilibrated Ni-sepharose (GE cat# 17371201) after which hSMOX was eluted using a 0–200 mM imidazole gradient. Fractions containing 410 nm absorption (FAD) were pooled and loaded onto a 5 ml CaptoQ Impres (GE cat# 17-5470-55) column pre-equilibrated with 10 mM HEPES pH 7.1 and eluted with a linear gradient of 0–0.5 M NaCl. Fractions containing hSMOX were pooled, and the buffer was exchanged into 10 mM HEPES, pH7.4, 150 mM NaCl using a Hiprep 26/10 desalting column (GE Cat#17-5087-01). Purified hSMOX was snap-frozen in liquid $N_2$ and stored at −80 °C until use.

Protein purity and size were quantified using chip-based electrophoresis (Bioanalyzer 2100, Agilent). All proteins employed in this study had a purity >95% and were consistent with expected MW. The FAD concentration was determined using UV/Vis spectroscopy employing the 450 nm absorption of the oxidized FAD group assuming $\varepsilon_{450nm} = 11,300$ $M^{-1}$ $cm^{-1}$. To calculate the relative ehSMOX FAD content ([FAD]/[SMO]×100%) the measured protein absorption at 280 nm was corrected for the contribution of FAD using the formula $Abs_{280,corr} = Abs_{280,exp} - 1.81 \times Abs_{450,exp}$ followed by calculation of [SMO] from $Abs_{280,corr}$ using the molar extinction coefficient calculated from the protein sequence. Note that the measured FAD content must be regarded as a lower limit because small impurities might contribute to the UV 280 nm absorption. For routine quantification of hSMOX and engineered constructs we used FAD absorption, as it is less prone to interference by protein or DNA impurities. Constructs employed in crystallization studies had purity >98% and a FAD content close to 100%.

**hPAOX expression and purification**. N-terminal 6xhistidine tagged hPAOX, isoform 1 (Q6QHF9-2; Refseq NP_690875.1) was expressed in baculovirus-infected Sf9 insect cells under the AcMNPV polyhedrin promoter (pPolh) transcriptional control. Codon-optimized cDNA was subcloned into pVL1393 (Expression Systems, 91–012) and co-transfected into Sf9 cells with linearized BestBac™ Baculovirus DNA (Expression Systems 91–002). P2 amplified virus was used to infect Sf9 cells for large-scale production in ESF 921 media (Expression Systems, 96-001) for 57 h to final viability of 85% and cell diameter of 21.3 μm.

6His-hPAOX protein was purified by IMAC. Homogenized lysate (10 ml per 1 g cell paste) in 25 mM HEPES, 500 mM NaCl, 20 mM imidazole, 250U/μl Benzonase (Novagen 71205), and 1x protease cOmplete inhibitor cocktail (Roche 05056489001) was bound to HisPur Ni-NTA resin (ThermoFisher) and eluted stepwise in loading buffer supplemented with 40, 80, 160, and 400 mM imidazole.

Pooled fractions from 10 and 40 mM imidazole elutions were yellow, indicating bound FAD co-factor. Pooled Ni-NTA fractions were concentrated with Jumbosep™ 10 K MWCO filter (Pall, OD010C65) at 10 °C and loaded onto a Superdex 200 26/600 sizing column (MilliporeSigma) equilibrated with 10 mM HEPES, pH 7.4, 150 mM NaCl. Eluted fractions were pooled and concentrated to 1.47 mg/ml and snap-frozen in liquid nitrogen for long-term storage at −80 °C.

**hSMOX,hPAOX and LSD1 activity assays**

*Reagents*. Pluronic F-127, chlorhexidine, albumin from chicken egg white, spermine dihydrate, $N^1$-acetylspermine trihydrochloride, potassium chloride, hydrogen peroxide, LSD1 Inhibitor IV, and methyl sulfoxide (DMSO) were purchased from MilliporeSigma (St. Louis, MO). 4-(2-hydroxyethyl)-1-piperazineethanesulfonic acid (HEPES) and ethylene glycol-bis (β-aminoethyl ether)-N,N,N′,N′-tetraacetic acid (EGTA) were purchased from Teknova (Hollister, CA) and Boston BioProducts (Ashland, MA), respectively. HyPerBlu was purchased from Lumigen (Southfield, MI), H3K4me2 peptide (residues 1–21; $H_2N$-ARTK(Me2) QTARKSTGGKAPRKQLA-OH) was synthesized by New England Peptide (Gardner, MA). Sodium chloride was purchased from VWR International (Radnor, PA) and human recombinant cleaved LSD1 (aa 158–852) was purchased from BPS Bioscience (San Diego, CA).

*hSMOX, hPAOX, and LSD1 enzyme assays*. The activity of hSMOX, hPAOX, and LSD1 enzymes was measured using the HyPerBlu chemiluminescent reagent for the direct detection of the hydrogen peroxide product. Spm, N-AcSpm, and H3K4me2 peptides were used as substrates for the hSMOX, hPAOX, and LSD1 assays, respectively. All hSMOX and hPAOX reactions were carried out in buffer A consisting of 10 mM HEPES, pH 7.4, 150 mM NaCl, 1 mM EGTA, 0.01% Pluronic F-127 and 0.05% ovalbumin unless otherwise stated. All LSD1 reactions were carried out in buffer B consisting of 50 mM Hepes, pH 7.5, 50 mM KCl, 0.01% pluronic F-127, and 0.01% ovalbumin. Hydrogen peroxide product formation was monitored 30 min after the addition of HyPerBlu on a PHERAstar microplate reader (BMG LABTECH, Cary, NC). All compounds were spotted into the assay plates using Echo acoustic dispensing technology (BeckmanCoulter, Indianapolis, IN). Unless otherwise stated, all reactions were carried out in 384-well assay format.

*$IC_{50}$ determination with hSMOX, hPAOX, and LSD1*. For the $IC_{50}$ determination of hSMOX, hPAOX, and LSD1, 50 nL/well of the JNJ-1289 or chlorhexidine serial

dilutions prepared in 100% DMSO was spotted into a 1536-well assay plate. 2 μL/well of 2x substrate was dispensed into the assay plate followed by 2 μL/well of 2x enzyme to initiate the reactions. The final concentration of hSMOX, hPAOX, and LSD1 enzyme in the assay was 2.5 nM, 60 pM, and 10 nM, respectively. The final concentration of Spm, N-AcSpm, and the H3K4me2 peptide was 30, 20 and 8 μM, respectively. After a 1 h incubation at room temperature, 4 μL/well of HyPerBlu only or 4 μL/well of HyPerBlu in the presence of 25 μM final chlorhexidine was added for quench and detection of the hSMOX and LSD1 versus the hPAOX reactions, respectively. The data were normalized to % inhibition using DMSO or 25 μM chlorhexidine to represent the 0 vs. 100% inhibited control reactions, respectively, for the hSMOX and hPAOx reactions. For LSD1, the data were normalized to % inhibition observed using DMSO or 10 μM LSD1 Inhibitor IV to represent the 0 vs. 100% inhibited control reactions, respectively. The data were fit to the 4-parameter $IC_{50}$ equation in GraphPad Prism.

*Determination of time dependent $IC_{50}$ shift with hSMOX and hPAOX.* To establish if time-dependent inhibition was observed, the inhibitor potency was determined plus and minus an E·I preincubation. 50 nL/well of the twofold inhibitor titrations in 100% DMSO were dispensed into the assay plate followed by the addition of 2.5 μL of 2x hSMOX or 2x hPAOX. A 2 h or 4 h E·I preincubation was carried out at a final concentration of 2 nM or 3 nM hSMOX or 60 pM hPAOX. Reactions were initiated with 2.5 μL of 2x Spm or 2x N-AcSpm substrate at a final concentration of 30 μM and 20 μM, respectively, and allowed to incubate for 1 h at room temperature. For the 0 h preincubation control, the addition of the substrate solution was made prior to the enzyme solution. The addition of 5 μL/well HyPerBlu reagent in the absence or presence of 25 μM chlorhexidine final was added for quench and detection of hydrogen peroxide product for the hSMOX and hPAOX reactions, respectively. The data were normalized to % inhibition observed with each preincubation time using DMSO or 25 μM chlorhexidine to represent the 0 vs. 100% inhibited control reactions, respectively. The data were fit to the 4-parameter $IC_{50}$ equation in GraphPad Prism.

*Determination of hSMOX, hPAOX, and LSD1 enzyme activity.* The enzyme activity of hSMOX, hPAOX, and LSD1 was measured by dispensing 2.5 μL/well of 2x enzyme into the assay wells followed by 2.5 μL/well of 2x substrate. The hSMOX reactions were evaluated at a final concentration of 0 to 25 nM enzyme in the presence of 30 μM Spm. The hPAOX reactions were evaluated at a final concentration of 0 nM to 0.76 nM enzyme at 10 μM N-AcSpm. The LSD1 reactions were evaluated at a final concentration of 0 nM to 100 nM enzyme at 50 μM H3K4me2 peptide. The addition of 5 μL/well HyPerBlu reagent in the absence or presence of 25 μM chlorhexidine final was added for quench and detection of hydrogen peroxide product for the hSMOX and LSD1 versus the hPAOx reactions, respectively. Product formation was measured at varying time points up to 1 h. The reaction rates were determined from the linear portion of each progress curve. A replot of the rate (μM/min) versus enzyme concentration demonstrated a linear relationship up to 12.5, 0.76, and 100 nM for the hSMOX, hPAOX and LSD1 reactions, respectively.

*Substrate kinetic parameters.* Determination of the $K_m$ for Spm, N-AcSpm or the H3K4me2 peptide was made at 2.5 nM hSMOX, 0.076 nM hPAOX and 20 nM LSD1. Briefly, 2.5 μL/well of the twofold substrate solutions prepared in buffer was added to the assay wells followed by 2.5 μL/well of the 2x enzyme to initiate the reactions. A time-course analysis was monitored up to 1 h at room temperature by quenching the reactions with 5 μL/well HyPerBlu reagent in the absence or presence of 25 μM chlorhexidine final for the hSMOX and LSD1 versus the hPAOX reactions, respectively.

The initial velocity data were fit to the Michaelis-Menten equation in GraphPad Prism.

$$Y = V_{max} * X/(K_m + X)$$

where $V_{max}$ is the maximal velocity, $X$ is the substrate concentration and $K_m$ is the Michaelis-Menten constant. The $k_{cat}$ value was determined by dividing $V_{max}$ value by the enzyme concentration.

The $K_m$ of Spm, N-AcSpm and the H3K4me2 peptide was determined to be 33.9 ± 2.7 μM, 20.1 ± 0.63 μM and 3.8 ± 0.35 μM for the hSMOX, hPAOX and LSD1 reactions, respectively.

*On rate determination of JNJ-1289 with hSMOX by progress curve analysis.* Sixty nL of JNJ-1289 serial dilutions prepared twofold (0–5000 nM) in DMSO at 100x final concentration were dispensed into the assay plate followed by the addition of 3 μL/ well of 2x Spm. Reactions were initiated with 3 μL/well of 2x hSMOX and quenched at several time points up to 1 h with 3 μL/well of 25 μM final chlorhexidine. All reactions were carried out in 50 mM Hepes, pH 7.4, 150 mM NaCl, 1 mM EGTA, 0.01% Pluronic F-127 and 0.05% ovalbumin at a final concentration of 60 μM Spm and 2 nM hSMOX. Thereafter, 9 μL/well of HyPerBlu reagent was added. The values of $K_i^{app}$, $k_{on}$ for a two-step time-dependent inhibition mechanism and calculated $k_{off}$ were determined from the fits to be 1.4 μM, $2.5 \times 10^3 M^{-1} s^{-1}$ and $2.5 \times 10^{-5} s^{-1}$, respectively. The $t_{1/2}$ dissociation value of 462 min was determined using the equation $0.693/k_{off}$.

*Mode of inhibition of JNJ-1289 versus Spm with hSMOX.* Sixty nL of 100x JNJ-1289 (312 nM final) was dispensed into the assay plate followed by the addition of 3 μL/ well of 2x Spm twofold serial dilutions and initiated with 3 μL/well of 2x hSMOX. All reactions were carried out in 50 mM Hepes, pH 7.4, 150 mM NaCl, 1 mM EGTA, 0.01% Pluronic F-127 and 0.05% ovalbumin at a final concentration of 7.8–1000 nM Spm and 2 nM hSMOX. Time point data up to 1 h of reaction time was captured by dispensing 3 μL/well of 25 μM final chlorhexidine to quench the reactions followed by the addition of 9 μL/well HyPerBlu reagent. The $k_{obs}$ values at varying Spm were fit to a competitive model for time-dependent inhibition.

**Thermal shift assay.** The fluorescent dye ANS (Molecular Probes) was used to monitor the thermal denaturation of hSMOX. Assay buffer consisting of 10 mM HEPES pH 7.5, 150 mM NaCl, 0.005% P20 was used for all dilutions. The assay was performed in 384-well Hard Shell thin-wall PCR plates (Bio-Rad). 120 nL of ligand was dispensed into the 384-well plate using acoustic dispensing technology followed by the addition of 4 μL of 0.2 mg/ml of hSMOX with 60 μM ANS dye. Replicates of 4% DMSO protein control were set up in each plate. Wells were covered with 1 μL silicone oil. Plates were centrifuged at 1000 rpm for 2 min. The Tm's were measured using a ThermoFluor instrument. The plates were heated from 30 to 80 °C at a rate of 1 °C per minute. The ΔTm is calculated as the difference between the Tm of each well and that of the average 4% DMSO protein control.

**Hydrogen deuterium-MS.** Sequence coverage for hSMOX was obtained from undeuterated control as follows: 5 μL of 46.5 μM hSMOX diluted in 15 μL Buffer D (50 mM HEPES pH 7.4, 150 mM NaCl) and added to 30 μL of ice-cold Buffer Q (1.2 M urea, 0.48% formic acid and 12 mM TCEP). Samples for ligand-bound states were prepared by mixing 1:100 compound: protein (mol/mol, 50 mM compound, 46.5 μM hSMOX stock) and incubated at room temperature for 15 min. The HDX-MS workflow for deuterated samples differed in the composition of Buffer D which was prepared in $D_2O$ and the reaction was stopped at 3 time points (10, 100, and 1000 s) in ice-cold Buffer Q and flash-frozen. LC-MS was performed using Nano LC-MS/MS (Dionex Ultimate 3000 RLSCnano System, ThermoFisher) interfaced with QE HF (ThermoFisher). Samples were thawed and pushed through protease type XVIII/pepsin column (NovaBioassays) and onto trap column (self-packed Poros R10, 2.1X4 cm) with 0.2% formic acid at flowrate 0.05 ml/min for 6 min using a syringe pump (Pump 11 Elite, Harvard Apparatus). Buffers, columns, lines and sample loop were all kept on ice as much as possible. After 6 min, the trap column was switched to be in line with analytical column (BioZen 2.6 μm, peptide XB-C18, LC column (50 × 2.1 mm, Phenomenex). A solvent gradient of 0–3 min: 2%B, 3–15 min: from 2% B to 15% B, 15–28 min: 15–30%B, from 28 to 33 min: 30–40%B (Buffer A: 0.2% formic acid, Buffer B: 0.1% formic acid in acetonitrile) was utilized to separate digested peptides. Mass was measured with a resolution of 120,000 and mass range from 300 to 2000 Da. The top 20 peaks were fragmented by HCD (relative collision energy 27%) and scanned with resolution 30,000 with dynamic exclusion for 30 s. Mgf file for sequencing runs was generated using Proteome discoverer 2.1 and searched against a custom database with the addition of common lab contaminants using Mascot v2.6. MS window was set at +/−10 ppm, MS/MS window set at +/−20 ppm. N-terminal acetylation was added as fixed modification and the enzyme was set as non-specific. When necessary (missing coverage), the peptide results were supplemented with X!Tandem search results using equivalent parameters (GPM Furry v3, theGPM.org). HDX time point results were analyzed using HDExaminer 2.5 (Sierra Analytics) with manual inspection.

**Crystallization.** The purified ehSMOX at 16 mg/ml was preincubated with 2 mM MDL72527 and used in crystallization trials employing a standard screen with ~1200 different crystallization conditions. Conditions initially obtained have been optimized using standard strategies, systematically varying parameters critically influencing crystallization, such as temperature, protein concentration, drop ratio, and others. The final crystallization conditions were 30% (v/v) MPD, 50 mM MES pH 5.5, 10 mM sodium acetate and 60 mM sodium fluoride. This solution already had cryo-protectant properties. Crystals have been flash-frozen and measured at a temperature of 100 K.

Crystals for ehSMOX bound to JNJ-1289 were obtained under the same crystallization conditions as above. Ligand JNJ-1289 was soaked into the crystals at a final concentration of 10 mM for 3 h.

*Data collection and processing.* All X-ray diffraction data for the here reported structures of ehSMOX have been collected at the PXII at SWISS LIGHT SOURCE (SLS, Villigen, Switzerland) using cryogenic conditions.

The crystals belong to space group P 3 2 1. Data were processed using the programs autoPROC, XDS[56,57], and autoPROC, AIMLESS. The phase information necessary to determine and analyze the structure was obtained by molecular replacement. The published structure of murine hPAOX (PDB ID:5MBX) was used as a search mode for ehSMOX structure, and this was subsequently used to solve the structure of ehSMOX bound to JNJ-1289.

Subsequent model building and refinement was performed according to standard protocols with COOT[58] and REFMAC[59], respectively. Detailed data

collection and refinement statistics for ehSMOX and ehSMOX bound to JNJ-1289 can be found in Table 1.

**NanoDSF**. All nanoDSF experiments were performed on a nanoTEMPER Prometheus NT.48 instrument. 10 μM purified SMOX was incubated with 100 μM compound of interest for 10 min and loaded onto the instrument. Thermal denaturation (25–90 °C) was conducted at 1.0 °C/min and 45% intensity. Tm was calculated using PR.ThermControl software as the inflection point of the first derivative of Trp fluorescence during the temperature gradient measured at 350/330 nm. All experiments were performed in triplicate.

**Conformational analysis by NMR**. All NMR spectra were recorded at 298 K in DMSO-$d_6$ on a Bruker 500 MHz instrument equipped with a 5 mm room temperature SmartProbe. Chemical shifts (δ values) are given in parts per million (ppm) and referenced to the DMSO (2.50 ppm) residual signal. For the conformational analysis of the molecule the following spectra were acquired: $^1$D $^1$H, 2D COSY, $^{13}$C-HSQC, $^{13}$C-HMBC and 2D EASY-ROESY (mixing time 300 ms; relaxation delay 5 s) using the standard pulse sequences available in TopSpin (v. 4.1.0, Bruker GmbH). NMR data were processed and analyzed using MestReNova (v. 14.2.1, Mestrelab Research S.L.). 2D cross-peaks from the 2D EASY-ROESY spectrum were integrated and converted into distances using the Stereofitter plug-in (v. 1.1.1.) embedded in MNova, where the PANIC method is used to normalize intensities relative to the diagonal peaks[60], and correction factors applied to compensate for the number of spins in each environment (corrected integral). A pair of adjacent aromatic protons situated at a constant distance independently of the conformation was selected as reference to calibrate the rest of the internuclear distances in the molecule.

The exploration of the conformational landscape of the molecule was carried out using MOE (v. 2020.09, CCG) using Low Mode Molecular Mechanics with the Amber10:EHT force field. The energy threshold was set to 7 Kcal/mol, and the cutoff for maximum atom deviation to 0.25 Å, generating 14 conformers. Calculated H-H distances from the conformers were matched with the averaged internuclear distances from the NOE data and those conformers that fitted were considered as representatives of the solution conformation.

**High mass resolution analyses**. High Mass resolution analyses were performed using a Xevo G2-S QTOF MS (Waters®, Milford, MA, USA) coupled to an IClass UPLC® (Waters®, Milford, MA, USA) system consisting of a binary pump with degasser, autosampler, thermostated column compartment and diode array detector. The MS was operated with an API-ESI source.

Spectra were acquired in positive mode. The capillary voltage was set to 0.25 kV. The cone voltage was set to 25 V. The source temperature was maintained at 140 °C. Acquisition mass range was $m/z$ 50–1200. Nitrogen was used as the nebulizer gas and argon as collision gas. Standard reversed phase gradients were carried out following the LC conditions detailed below. The standard injection volume was 1 μL.

A Lockmass device with Leucine-Enkephalin as standard substance was used for mass calibration. Data acquisition was performed with MassLynx™/OpenLynx™ 4.1 software (Waters®, Milford, MA, USA).

*LC2 method*. Reversed phase UPLC was carried out on a BEH C18 column (1.7 μm, 2.1 × 50 mm) from Waters, with a flow rate of 1.0 mL/min, at 50 °C. The gradient conditions used are: 95% A (NH4OAc, 6.5 mM in H2O + 5% CH3CN), 5% B (CH3CN), to 5% A in 4.6 min, held for 0.4 min, with a total run time of 5 min. Acquisition range for the UV-PDA detector was set to 200–450 nm. MS (QTOF): see general description.

**CETSA experiments A549 cells**. A549 (adenocarcinoma human alveolar basal epithelial cells) express relatively high levels of hSMOX. Their use has been widely reported in the literature for the investigation of the polyamine catabolic pathway[17]. In our case, using hSMOX high-expressing cells with CETSA and AlphaLISA systems provided the optimal platform for screening promising small molecules for hSMOX inhibition.

*Cell preparation*. T175 culture flasks with 80–90% confluent cells were used for harvesting after washing two times with PBS followed by the addition of 2 mL TrypLE™select (Gibco) per flask and incubation for 5 min at 37 °C. After dissociation, cells were resuspended in 1 mL culture medium (DMEM (Gibco) with 10% fetal calf serum) and 7 mL PBS. Cells from multiple flasks were pooled, spun down for 5 min at $300 \times g$, and resuspended to 20 million cells/mL in PBS.

*Compound and temperature treatment*. Compounds were dissolved in DMSO to 1 mM stocks. Compound stocks and DMSO controls were added to vials containing 1 mL cell suspension to an end concentration of 10 μM compound and 1% DMSO. Vials were incubated at 37 °C for 1 h while shaking at 125 rpm. Subsequently, vials were cooled on ice and cells were distributed to PCR vials at 20 μL per vial. Transient heating of the cells was performed using a PCR Thermocycler. Vials were heated up to 54 °C for 1 min and then immediately cooled on ice. For

temperature range experiments the heating step was repeated increasing the temperature with each next vial with 1 °C. After the cooling step cells were lysed by adding 80 μL 1x AlphaLISA-buffer (Perkin Elmer).

*AlphaLISA*. The PerkinElmer's AlphaLISA is a highly selective and sensitive chemiluminescent, no-wash assay that allows for the distinction of intact from denatured hSMOX through the disappearance of the conformational epitopes. The hSMOX specific AlphaLISA was developed using internally monoclonal rabbit antibody mAb#2 and F(ab')$_2$ fragment Fab#33 against hSMOX containing a C-terminal His-tag. Both mAb#2 and Fab#33 are specific for hSMOX and selected for their ability to detect hSMOX without interfering with its catalytic activity. Care was taken to confirm that both mAb#2 and Fab#33 bind to distinct (conformational) epitopes. Both mAb#2 and Fab#33 were shown to bind recombinant hSMOX without any sign of competition (manuscript in preparation). The Alpha donor beads (Perkin Elmer) are coupled with conjugated goat polyclonal F(ab')$_2$ fragment targeting the Fc region of rabbit IgG-mAb#2 (but not the rabbit Fab#33 molecules lacking the Fc region). The anti-6X His acceptor beads (Perkin Elmer) bind exclusively to the His-tagged Fab#33. mAb#2/ Fab#33 mixture was prepared by diluting mAb#2 to 0.5 nM and Fab#33 to 5 nM in 1x Alpha-buffer. 12.5 μl mAb#2/ Fab#33 mixture was added to 25 μL lysate and incubated for 1 h at RT. Then 12.5 μL acceptor beads (50 μg/mL) were added followed by an additional incubation of 1 h at RT. For the last step of the assay 20 μL was transferred in duplicate to a 384 well white assay plate (OptiPlate-384, Perkin Elmer) followed by the addition of 5 μL donor beads (200 μg/mL) per well. After 1 h Alpha signal was read using an Alpha compatible reader.

**Reporting summary**. Further information on research design is available in the Nature Research Reporting Summary linked to this article.

## Data availability

The datasets generated and/or analyzed during the current study are available in the supplementary material and from the corresponding author on reasonable request. Regarding the crystal structures, protein coordinates and additional data are available at RCSB Protein Data Bank with accession codes PDB ID: 7OXL and PDB ID: 7OY0 for ehSMOX in complex with MDL72527 and ehSMOX in complex with JNJ-1289, respectively.

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

## Acknowledgements

We thank Kurtis Bachman, Murray McKinnon, Paul Klatser and Els Brinkman (from *The Janssen Pharmaceutical Companies of Johnson & Johnson*) for supporting this research, and Sven Johannsson, Stephan Krapp and Aaron Alt from *Proteros biostructures GmbH* for supporting with protein expression, protein crystallization, and for helpful discussions.

## Author contributions

All authors have agreed to their contribution to the work: E.D.: biochemical development of hSMOx HTS assay and hPAOX and LSD1 selectivity assays/characterization of JNJ-1289 and chlorhexidine with hSMOX and hPAOX and contributing to the manuscript preparation. S.A.: design, development execution and analysis of HDX-MS experiments/data and contributing to the manuscript preparation. A.T.: established the protein expression and purification and characterization procedures supporting ehSMOX candidate screening selection. Initially developed of the wt hSMOX purifications and activity assay and contributed to the manuscript preparation. D.R.: High-throughput screening campaign to identify inhibitor, performed the experiments. R.O.M.: High-throughput screening campaign experimental design and contributed to the manuscript preparation. D.K.: contributed to the biochemical assay development and to the manuscript preparation. C.B.: computational chemistry support and core team member, including a selection of hits for confirmation of the HTS, data preparation and analysis of HTS hits. Analogue searching around interesting clusters. Analysis of the crystal structures, gaining structural insights and explaining SAR and performing docking experiments. Contributing to the manuscript preparation. C.M.L.: analysis of hSMOX HTS hits and selection of clusters for further validation selection of JNJ-1289 as candidate for crystallography. J.L.: coordinate the MedChem strategy for the project and contribute the manuscript preparation. L.F.: synthesis

of hit compound JNJ-1289, which validated the hit in several assays and used in biology studies. JNJ-1289 ADME profiling, analogue searching around the hit to understand SAR and follow up MedChem strategy for the cluster. B.D.: developed and executed the TSA assay. J.K.: contributed to development of CETSA assay and to the manuscript. V.K.: developed and executed CETSA assay and contributed to the manuscript. S.F.: developed and executed CETSA assay and contributed to the manuscript. P.S.: interpretation of structural biology results and contributed to the manuscript. R.K.: expression and pur- ification of hPAOX and contribution to the manuscript. R.C.: free ligand conformational studies by NMR and contribution to the manuscript. M.T.: crystallized and analysed the results of the ehSMOX structure in complex with JNJ-1289 and contributed to the manuscript. A.I.: engineered the protein and designed the experiment for the hSMOX candidate selection and analyses of the crystal structures features. Directed the overall project and wrote the manuscript with input from all other authors.

## Competing interests
The authors declare no competing interests.
