## [Peer Review File · Communications Biology]

Reviewers' comments:

Reviewer #1 (Remarks to the Author):

The paper describes the first structure of SMOX along with hit finding and characterization of a hit, JNJ-1289. The structure of SMOX has been elusive with many groups failing to achieve it over the years. The protein engineering section presented here is very elegant and systematic. It is clearly presented and the resulting structure is of great general interest in the field of polyamine catabolism. The hit finding campaign and subsequent characterization is also very clear. A few concerns/comments:

The lack of electron density for MDL72527 is compared to the mPAOX-MDL72527 (Line 217). Here the description is not correct. In the mPAOX structure there is clear electron density for MDL72527 up until N5, which is involved in hydrogen interaction. I'm guessing the authors are mixing up N5 of MDL72527 with N5 of the isoalloxazine ring, due to the unfortunate coincidence that the atom names are the same! I would like to see a supplementary figure showing the electron density extending from the isoalloxazine N5 atom.

Supplementary figures 4 and 5 describe differences and similarities between SMOX and PAOX, which are illustrated by two sets of side-by-side comparisons. These figures are very difficult to interpret. Could the authors consider 2 single panel figures with the structures overlaid instead?

A weak point of the paper is the fact that crystals could only be achieved in the presence of MDL72527. The co-crystal structure with JNJ-1289 is therefore a complex with not only one, but two inhibitors. The authors have used HDX-MS to map the ligand binding site, however, the resolution of these studies can't completely rule out the possibility that the observed binding site is influenced by MDL72527, in fact the areas which show as protected by MDL72527 (Fig3b) and JNJ-1289 (Fig 6b) seem to be nearly identical. Can the authors provide additional evidence that simultaneous binding of MDL72527 and JNJ-1289 is possible in solution, by e.g ITC?

Reviewer #2 (Remarks to the Author):

In this study, the authors reported structure of human spermine oxidase (hSMOX) in complex with a highly selective inhibitor. hSMOX plays an important role in polyamine catabolism and is a therapeutic target in inflammation and cancer but is difficult to be crystallized. The authors used systematic, rational engineering approaches to successfully co-crystallize hSMOX with its inhibitor MDL75257 and solved their complex structure. As MDL75257 has poor selectivity and low potency, the authors identified a potent and selective inhibitor JNJ-1289 using a high-throughput enzymatic activity assay and determined the co-crystal structure of hSMOX with JNJ-1289. Altogether the results revealed the structural basis of selective inhibition of hSMOX by JNJ-1289 and enabled the further development of selective hSMOX inhibitors.

The experiments in this study are well done and the results are convincing. However, there are some issues that require clarification or corrections before this manuscript can be accepted for publication.

Specific comments:

(1) Page 12. The statement "...suggesting that JNJ-1289 is a competitive hSMOX with respect to Spm" is not consistent with the structural analysis. The structure showed that JNJ-1289 binds to an allosteric site of hSMOX and the molecular modeling analysis showed that Spm and JNJ-1289 can be accommodated at the same time, strongly suggesting that JNJ-1289 is an allosteric inhibitor rather than a competitive Inhibitor. If JNJ-1289 is a competitive Inhibitor, the binding of JNJ-1289 and Spm to hSMOX would be mutually exclusive.

(2) The inhibition of hSMOX by JNJ-1289 was reported to be time-dependent but only two time points were used for the analysis. More time-based data points should be used to justify the conclusion. Will longer preincubation time effectively increase the binding of the inhibitor and enhance the inhibition effect?

(3) JNJ-1289 has good in vitro cellular permeability but CETSA showed no cellular hSMOX engagement. The authors stated that such failure may be caused by high cellular Spm

concentration. If this is the case, what is the cellular Spm concentration vs the concentration of JNJ-1289 used in CETSA ?

(4) Important parts referred in the text of the H/D exchange heat map in Supplementary Figure 1 must be highlighted for understanding of the experimental data.

(5) In Supplementary Fig. 14a. What is the ball-and stick model ? The bound JNJ-1289 and modelled Spm should be labelled.

(6) In the sentence, "To further inform our construct design..." in Page 5, "inform" must be corrected to "improve"

(7) In Supplementary Fig 9(c), horizontal axis: Spm concentration should be in nM rather than in μM . Also, it would be good to have the raw inhibition curves for Supplementary Fig 9(c) for intuitive visualization and understanding of the binding and inhibitory process.

Reviewer #3 (Remarks to the Author):

The manuscript by Elsie Diaz et al " Structure of human spermine oxidase in complex with a highly selective allosteric inhibitor " describes the first structure of SMOX enzyme, identification of novel specific inhibitor using biochemical and cellular assays and presents the structure of the enzyme with the novel inhibitor. The findings are significant and warrant publication.

There are several major problems with presentation and analysis of novel structure.

Major critique:

1. Structures can not be published without full refinement statistics for each deposited and discussed structure.

2. Illustration of electron density maps are critical for key structural elements, particularly for ligands and elements with questionable conformation.

Consequently, it is impossible to appreciate and validate structural features and hypothesis presented in the current manuscript.

3. There is no sequence comparison analysis and discussion of conserved structural elements as well as conservation of structural parts discussed in the paper and explanation of structural basis of previously explored mutants.

4. Ligand docking should be done using available programs and Spm binding should be analyzed in more details including discussion of conserved amino acids, preferably, comparison with other known complexes with Spm of other enzymes, and more detailed analysis of similarities and differences with PAOX. This will significantly increase the impact of novel structure.

5. There is no description of HTS experiment. Parameters critical for efficient HTS assay and description of libraries should be included.

6. Many structural figures need improvement as they are too confusing and do not properly illustrate discussion points.

7. The text should be carefully edited and improved in many places throughout the main text and figure legends.

Additional specific points:

1. While large part of the manuscript devoted to a multistep mutagenesis to obtain crystallizable mutant, there is no attempt to analyze how each of modifications can affect crystallization based on crystal packing.

2. Quantitative comparison of basic enzymatic parameters is required to properly compare

engineered enzyme variants. It is unclear why the authors decided to present qualitative Suppl. Fig 2 instead of actual numbers.

3. There should be at least a short discussion of potential influence (or lack of such) of mutagenesis on structural elements discussed in the manuscripts.

4. (L. 166) What is RMSD between SMOX and PAOX.

5. (L. 185) Loop length and lack of homology are not valid reasons not to build a model of the loop, particularly, since this loop was modeled in the second lower resolution structure. The author should state that there is no density for this part. However, if density is visible, even partially, it is worth building as much as possible.

6. (L.197) Are all of those pi-interactions or hydrophobic interactions? If they form a hydrophobic core between helices, then are there other hydrophobic residues which can stabilize the hydrophobic core instead of missing aromatic residues?

7. (L. 202) Why should a disulfide bond break a-helix?

8. (L. 209) There is no clear illustration of how far Thr substituting Asn313 is from the ligand and why it should correspond to Asn313? Again, sequence alignment and a better illustration of structural superposition should be presented.

9. Fig. 4B – Such representations with a nontransparent surface of one protein and tiny features protruding out of this surface from another protein is not very informative. Please, try to make a better figure to illustrate structural comparison.

10. (L.212-214) There is no discussion of how different residues will affect substrate preference, particularly, since Spm and N-AcSpm have similar structural parts.

11. (L. 215) Was MDL72527 present in the crystallization of WT and other mutants.

12. (L. 215-216) Provide an electron density map calculated before modeling MDL72527 (or from a omit map with absent substrate) to illustrate modeling of MDL72527.

13. (L.224) Can His62 be modeled in two alternative conformations to support the hypothesis?

14. (L. 244) Substitute "campaign" for a more informative term.

15. Suppl Fig. 4: N-AcSpm molecule is modeled from superposition of two protein structures, not "overlapped".

16. (L. 256) Where did the number 67 μ M come from?

17. (L. 310) A figure with the electron density for JNJ-1289 should be included.

18. (L.314) 190-210 loop is not highlighted in Fig. 6A.

19. (L.331) There is no explanation of the coloring scheme and where loops of interest are in Suppl. Fig. 11.

20. (L.336-339) Which structural feature? Shape of the tunnel? It is difficult to relate Suppl Fig 12a with other structural presentations.

21. (L.351) How was the scheme on Fig. 7C created? Provide distances for critical bonds.

22. (L. 387-394) It will be more beneficial to use auto docking programs and then analyze results, rather than manual docking and refinement (undescribed).

23. (L.399) Are these residues conserved in SMOX and other polyamine oxidases?

24. (L. 407-412) SITE1 and SITE2 are mentioned but are not described in detail and not used for design of novel inhibitors. This paragraph is not informative and can be omitted.

25. (L. 420) See comment #11 above.

26. (L.424) "Asn313 ... occurs as Thr309": Substituted, replaced? See point #8 above.

27. (L.425) If the previous hypothesis about Glu196 and Ser198 is not supported by the structure, then how do these amino acids affect substrate specificity accordingly to new structural data? Or the previous biochemical data are wrong/erroneous?

28. (L.429) How was the surface charge calculated?

29. (L.440) "Spm was able to dock": Both Spm and JNJ-1289 can be docked simultaneously without steric clash?

"Structure of human spermine oxidase in complex with a highly selective allosteric inhibitor" tracking number: COMMSBIO-21-2025A for consideration as an Article in Communication Biology.

Responses to the reviewers:

Reviewer #1 (Remarks to the Author):

The paper describes the first structure of SMOX along with hit finding and characterization of a hit, JNJ-1289. The structure of SMOX has been elusive with many groups failing to achieve it over the years. The protein engineering section presented here is very elegant and systematic. It is clearly presented and the resulting structure is of great general interest in the field of polyamine catabolism. The hit finding campaign and subsequent characterization is also very clear.

A few concerns/comments:

1. The lack of electron density for MDL72527 is compared to the mPAOX-MDL72527 (Line 217). Here the description is not correct. In the mPAOX structure there is clear electron density for MDL72527 up until N5, which is involved in hydrogen interaction. I'm guessing the authors are mixing up N5 of MDL72527 with N5 of the isoalloxazine ring, due to the unfortunate coincidence that the atom names are the same! I would like to see a supplementary figure showing the electron density extending from the isoalloxazine N5 atom.

The reviewer is correct; this was an ambiguity of the naming conventions of the FAD and MDL72527. In the manuscript (line 222-253), we have clarified this and added a supplementary figure with the omit electron density for MDL72527 (Supplementary Fig. 6).

2. Supplementary Figures 4 and 5 describe differences and similarities between SMOX and PAOX, which are illustrated by two sets of side-by-side comparisons. These figures are very difficult to interpret. Could the authors consider 2 single panel figures with the structures overlaid instead?

Thank you for this advice. We made a new version of Figure 4 in the main article with the structures of SMOX and PAOX overlaid. In addition, Supplementary Fig. 5 has also been revised, and new figures have been added to the manuscript, hopefully creating more clarity to the overall paper.

3. A weak point of the paper is the fact that crystals could only be achieved in the presence of MDL72527. The co-crystal structure with JNJ-1289 is therefore a complex with not only one, but two inhibitors. The authors have used HDX-MS to map the ligand binding site, however, the resolution of these studies can't completely rule out the possibility that the observed binding site is influenced by MDL72527, in fact the areas which show as protected by MDL72527 (Fig3b) and JNJ-1289 (Fig 6b) seem to be nearly identical. Can the authors provide additional evidence that simultaneous binding of MDL72527 and JNJ-1289 is possible in solution, by e.g ITC?

We would like to thank the reviewer for the question. A point worth noting is that the reported HDX signature of SMOX with JNJ-1289 was generated in the absence of MDL-72527. While some of the areas displaying protection offered by MDL72527 and JNJ-1289 overlap, there are distinct differences in deuterium uptake kinetics for the two compounds. One such example can be seen in the region a.a.210-225 where we see a consistent and

sustained reduction in deuterium uptake consistent with the placement of MDL72527 associated modification of FAD. The same region in JNJ-1289 bound form shows only a modest level of protection that dissipates over time, consistent with crystal structure where the secondary structures in this region are more resolved (See figure below).

Due to company reorganization and changing of priorities, it was not possible to perform the ITC experiment as requested from the reviewer. However, we carried out a series of nanoDSF experiments (we to parse the differences in stabilization offered by the two ligands). The experiments show that both ligands increase the thermal stability of the protein independent of each other (See data summary below). Interestingly, the protein is most stable in presence of JNJ-1289 alone. nanoDSF experiments have been added in the manuscript (line 364; 719) and Supplementary Fig. 19.

Taken together, these data show that -

- JNJ-1289 can engage SMOx independent of MDL72527 in solution.
- Presence of MDL72527 does NOT affect the stabilization offered by JNJ-1289.

Reviewer #2 (Remarks to the Author):

In this study, the authors reported structure of human spermine oxidase (hSMOX) in complex with a highly selective inhibitor. hSMOX plays an important role in polyamine catabolism and is a therapeutic target in inflammation and cancer but is difficult to be crystallized. The authors used systematic, rational engineering approaches to successfully co-crystallize hSMOX with its inhibitor MDL75257 and solved their complex structure. As MDL75257 has poor selectivity and low potency, the authors identified a potent and selective inhibitor JNJ-1289 using a high-throughput enzymatic activity assay and determined the co-crystal structure of hSMOX with JNJ-1289. Altogether the results revealed the structural basis of selective inhibition of hSMOX by JNJ-1289 and enabled the further development of selective hSMOX inhibitors. The experiments in this study are well done and the results are convincing. However, there are some issues that require clarification or corrections before this manuscript can be accepted for publication. Specific comments:

1. Page 12. The statement “...suggesting that JNJ-1289 is a competitive hSMOX with respect to Spm” is not consistent with the structural analysis. The structure showed that JNJ-1289 binds to an allosteric site of hSMOX and the molecular modeling analysis showed that Spm and JNJ-1289 can be accommodated at the same time, strongly suggesting that JNJ-1289 is an allosteric inhibitor rather than a competitive Inhibitor. If JNJ-1289 is a competitive Inhibitor, the binding of JNJ-1289 and Spm to hSMOX would be mutually exclusive.

We want to thank the reviewer for pointing out this issue. Unfortunately, we have never been able to obtain a crystal structure of hSMOX in the presence of its substrate. Following known procedures to the experts in the field, hSMOX -JNJ1289 crystal was obtained by soaking the ligand into hSMOX-MDL72527 crystals. Although the structural model offers the possibility to accommodate both JNJ-1289 and Spm simultaneously virtually, this does not seem to occur in our binding analysis. Although competitive inhibitors often bind in the same binding site as the substrate, this is not a requirement. Indeed, the additional experiments analyzing the mode of inhibition of JNJ-1289 in the presence of a range of Spm concentrations (Supplementary Figure 11c and d), further confirms JNJ-1289 as competitive inhibitor binding to an allosteric site of the free enzyme, preventing substrate binding.

2. The inhibition of hSMOX by JNJ-1289 was reported to be time-dependent but only two time points were used for the analysis. More time-based data points should be used to justify the conclusion. Will longer pre-incubation time effectively increase the binding of the inhibitor and enhance the inhibition effect?

We would like to thank the reviewer for the question. We have provided additional data to measure the time-dependent inhibition of JNJ-1289 at three-time points (0h, 2h and 4h) with hSMOX enzyme. Further pre-incubation of JNJ-1289 with hSMOX for 4h resulted in potency within 2-fold the results observed using a 2h pre-incubation (Supplementary Fig 10). These data suggested that 2h was sufficient for the equilibrium binding of JNJ-1289 to hSMOX. We have added the following sentence on line 279 of the manuscript: “Further pre-incubation of JNJ-1289 with hSMOX for 240 min resulted in a potency within 2-fold the results observed using a 120 min pre-incubation (Supplementary Fig 10).”

These results are more qualitative and demonstrate that JNJ-1289 binds to hSMOX in a time-dependent manner. A quantitative treatment of time-dependent inhibition is shown in Supplementary Figure 9a, which measures the onset of inhibition of JNJ-1289.

3. JNJ-1289 has good in vitro cellular permeability but CETSA showed no cellular hSMOX engagement. The authors stated that such failure may be caused by high cellular Spm concentration. If this is the case, what is the cellular Spm concentration vs the concentration of JNJ-1289 used in CETSA?

The concentration of Spm and polyamines is generally considered in the millimolar concentration range, although the free polyamine pool represents only 7-10% of the total cellular polyamine content. Furthermore, free polyamine pools are highly regulated. In a deeper analysis, we also concluded that the hypothesis we reported in the previous version of the manuscript might not be correct as indeed some compounds as well as MDL75257 did show a positive signal in cell at the physiological concentration of Spm. We modified the manuscript accordingly (line 489). At this moment, we don't have a clear explanation for the inconsistent results of JNJ-1289. Additional experiments will be required which we consider outside the scope of the present manuscript.

4. Important parts referred in the text of the H/D exchange heat map in Supplementary Figure 1 must be highlighted for understanding of the experimental data.

Thank you for the suggestion. The figure has been modified as Supplementary Figure 2 to highlight areas and residues representing significant differences in deuterium uptake in the revised manuscript.

5. In Supplementary Fig. 14a. What is the ball-and stick model? The bound JNJ-1289 and modelled Spm should be labelled.

FAD and modelled Spm have been labelled. The figure is now Supplementary Fig.18

6. In the sentence, "To further inform our construct design..." in Page 5, "inform" must be corrected to "improve"

The change has been implemented (line 123)

7. In Supplementary Fig 9(c), horizontal axis: Spm concentration should be in nM rather than in μ M. Also, it would be good to have the raw inhibition curves for Supplementary Fig 9(c) for intuitive visualization and understanding of the binding and inhibitory process.

Thank you for pointing this out. We agree with this suggestion and have added the raw inhibition curves for Supplementary Fig 9(c), which has been updated to Supplementary Fig 11(d) including raw inhibition curves. The horizontal axis of Fig 9(d) was correct to suggest that the Spm concentration is in μ M. What was incorrect was the Spm concentration written underneath the figure legend, which now reads "Spm concentrations (7.8 – 1000 μ M instead of 7.8 – 1000 nM), adjusted in line 127 supplementary information.

Reviewer #3 (Remarks to the Author):

The manuscript by Elsie Diaz et al " Structure of human spermine oxidase in complex with a highly selective allosteric inhibitor " describes the first structure of SMOX enzyme, identification of novel specific inhibitor using biochemical and cellular assays and presents the structure of the enzyme with the novel inhibitor. The findings are significant and

warrant

publication.

There are several major problems with presentation and analysis of novel structure.

Major critique:

1. Structures cannot be published without full refinement statistics for each deposited and discussed structure.

We apologize for this omission. Data collection and refinement statistics for both structures have been placed in Supplementary Table 1.

2. Illustration of electron density maps are critical for key structural elements, particularly for ligands and elements with questionable conformation. Consequently, it is impossible to appreciate and validate structural features and hypothesis presented in the current manuscript.

Thank you for this comment. Electron density for MDL72527 (Supplementary Figure 6) and JNJ-1289 (Supplementary Figure 13) have been added.

3. There is no sequence comparison analysis and discussion of conserved structural elements as well as conservation of structural parts discussed in the paper and explanation of structural basis of previously explored mutants.

We appreciate the reviewer comment. We have addressed this comment in the revised manuscript showing in Supplementary Figure 1 the alignment of the different engineered constructs and the structural elements introduced during the iteration process. At line 116 of the manuscript, we also have addressed this concern.

4. Ligand docking should be done using available programs and Spm binding should be analyzed in more details including discussion of conserved amino acids, preferably, comparison with other known complexes with Spm of other enzymes, and more detailed analysis of similarities and differences with PAOX. This will significantly increase the impact of novel structure.

Default docking with MOE was tried but the results were not satisfactory to the authors. All manipulations and steps are performed in the computational program MOE. Instead of using randomly generated poses or even constrained docking and scoring as done by the default docking procedure, we preferred a more user-controlled (manual) ensure that interactions with conserved residues are kept in the proposed fitting hypothesis and others interactions are being optimized by iterative constraint minimization procedures. We believe that this is the best way to create a realistic docking hypothesis, nevertheless are also aware that there can be other binding hypotheses with such a flexible molecule. On the other side, there is also no guarantee that the crystal structure, even after correcting for some transformations as described, would have all the amino acids properly positioned to create the true binding pose.

5. There is no description of HTS experiment. Parameters critical for efficient HTS assay and description of libraries should be included.

More information regarding the HTS experiment, format and assay statistics for this effort has been added (line 265). A specific description of chemical libraries screened is beyond the scope of this manuscript and not customary in the field.

6. Many structural figures need improvement as they are too confusing and do not properly

illustrate discussion points.

We would like to thank the reviewer for this advice. Structural figures of the main article and of the supplementary information have been extensively revised.

7. The text should be carefully edited and improved in many places throughout the main text and figure legends.

We apologize for the errors, and we have carefully edited the manuscript as suggested from the reviewer to improve readability.

Additional specific points:

8. While large part of the manuscript devoted to a multistep mutagenesis to obtain crystallizable mutant, there is no attempt to analyze how each of modifications can affect crystallization based on crystal packing.

In our analysis of the crystal packing none of the mutations introduced seems to create crystal contacts that may clearly contribute to the crystal packing. Therefore, we deduce that the contribution of all the engineered features is what contributes to an increased stability of the protein and its increased crystallization propensity. Furthermore, the crystal could only be obtained in the presence of the ligand MDL (line 220) underline the crucial role of this additional stabilization. Furthermore, additional attempt to crystallize wild-type hSMOX with several monoclonal antibodies specifically selected towards diverse hSMOX epitope have also failed (line 450).

9. Quantitative comparison of basic enzymatic parameters is required to properly compare engineered enzyme variants. It is unclear why the authors decided to present qualitative Suppl. Fig 2 instead actual numbers.

Thank you for pointing this out. We have added the complete analysis of the kinetic parameters of the engineered enzyme variants and have updated Supplementary Fig 3, in the revised version, with the results. The data demonstrate that the k_{cat}/K_m values of all the constructs were within two-fold of one another.

10. There should be at least short discussion of potential influence (or lack of such) of mutagenesis on structural elements discussed in the manuscripts.

The introduced mutations have been performed in protein segments deprived of structure to decrease the chance of impacting the protein's overall structure. Further clarifications are added in line 116 of the manuscript in Figure 2 and with the addition of Supplementary Figure 1.

11. (L. 166) What is RMSD between SMOX and PAOX.

We apologize for the missing information. The RMSD has been added to the manuscript (line 169 and 328)

12. (L. 185) Loop length and lack of homology are not valid reasons not to build a model of the loop, particularly, since this loop was modeled in the second lower resolution structure. The author should state that there is no density for this part. However, if density is visible, even partially, it is worth to build as much as possible.

The wording has been changed to reflect that these loops were not built because of a lack of electron density.

13. (L.197) Are all of those pi-interactions or hydrophobic interactions? If they form hydrophobic core between helices, then are there other hydrophobic residues which can stabilize hydrophobic core instead of missing aromatic residues?

As described in the manuscript, the authors have labeled several hydrophobic interactions as Pi-Pi interactions. Most pi interactions are hydrophobic interactions (set aside those with cations). An additional H-bond between E185 and Y123 is only present in ehSMOX (line 205).

14. (L. 202) Why disulfide bond should break α -helix?

The referee is correct as the phrase was not precise: the authors didn't mean that cysteine bridges generally break helical motif; we wanted to indicate that the Sa3 helix of hSMOX has a shorter helical motif than the latter, the same helix of mPAOX. This interruption occurs precisely at the cysteine bridge Cys187-Cys373, enabling a different loop position compared to the one of mPAOX: while in mPAOX the loop goes towards the back of the protein, in the case of hSMOX comes towards the front. The same loop position is also present in the structure of the bound form of hSMOX with JNJ-1289. We have clarified this statement in the manuscript (Figure 4 line 207 and 252).

15. (L. 209) There is no clear illustration of how far Thr substituting Asn313 is from ligand and why it should correspond to Asn313? Again, sequence alignment and better illustration of structural superposition should be presented.

Supplementary Figure 1 and 5 have been introduced to clarify this point.

16. Fig. 4B – Such representations with nontransparent surface of one protein and tiny features protruding out of this surface from another protein is not very informative. Please, try to make a better figures to illustrate structural comparison

Figures of the main article and supplementary information have been extensively revised, hopefully providing a better illustration of the structural features.

17. (L.212-214) There is no discussion of how different residues will affect substrate preference, particularly, since Spm and N-AcSpm have similar structural parts.

Please refer to the question below.

18. In supplementary Figure 5 we show how N-Ac SPM cannot be accommodated in hSMOX as in mPAOX (line 230 of the manuscript). This substrates are very flexible and it is possible that they can accommodate in different way.

The reviewer is correct, the high flexibility of the substrate as well as of the active site may allow different substrate positions. However, we don't have any data to support these possibilities in hSMOX, although NAcSpm is known to be a poor substrate of SMOX. Additionally, it is interesting the work of Keinänen et al (Biosci Rep 2018) in which it is shown how actually stereospecificity and regiospecificity of FAD-dependent polyamine oxidases could be controlled with small substrate modification. This could be consistent with our observation that the different loop orientation in heSMOX hampers the position

that NAcSpm adopts in mPAOX.

19. (L. 215) Was MDL72527 present in crystallization of WT and other mutants

Yes, MDL72527 presence appeared necessary for crystallization. In its absence, we have not been able to get a crystal with proper diffraction properties both with the wt hSMOX and the engineered variants (line 450 manuscript).

20. (L. 215-216) Provide an electron density map calculated before modeling MDL72527 (or from a omit map with absent substrate) to illustrate modeling of MDL72527.

Omit density for MDL72527 has been added as a Supplementary Figure 6.

21. (L.224) Can His62 be modeled in two alternative conformations to support hypothesis?

We have added a statement clarifying that there is insufficient electron density to model His62 in alternate conformations in the apo-structure and to validate this hypothesis (line 422).

22. (L 244) Substitute “campaign” for more informative term.

Campaign is a common phrase used to describe High-throughput screening efforts. However, the word campaign has been removed from the section heading (line 256) for clarity. Other changes made to the hSMOX inhibitor discovery section should prove more informative as to the effort undertaken to identify the inhibitor.

23. Suppl Fig. 4: N-AcSpm molecule is modeled from superposition of two protein structures, not “overlapped”.

The word has been substituted (line 53).

24. (L. 256) Where number 67 uM came from?

The reviewer is correct. We have corrected the value of 67 μM with $> 2 \mu\text{M}$ for the IC₅₀ of JNJ-1289 with LSD1 and hPAOX enzymes. “To measure the selectivity of JNJ-1289, we assayed the inhibitor against hPAOX and LSD1 and found that JNJ-1289 has IC₅₀ values greater than 2 μM against both enzymes (Fig. 5b).” (line 268)

25. (L. 310) A figure with the electron density for JNJ-1289 should be included.

Supplementary Figure 13 with omit and 2Fo-Fc electron density for JNJ-1289 has been added.

26. (L.314) 190-210 loop is not highlighted in Fig. 6A.

The loop is now highlighted in Figure 6a and Figure 7a.

27. (L.331) There is no explanation of coloring scheme and where loops of interest are in Suppl. Fig. 11.

The coloring scheme explanation has been added in the figure legend (now Supplementary Figure 14) and a further clarification of the figure.

28. (L.336-339) Which structural feature? Shape of the tunnel? It is difficult to relate Suppl Fig 12a with other structural presentations.

The figure illustrations and legends of Supplementary Figure 14 and 15 have been revised to clarify further what the authors mean by the shape of the tunnel.

29. (L.351) How scheme on Fig. 7C was created? Provide distances for critical bonds.

This graph is produced by the ligand interactions tool from MOE. There is an option to add distances to the plot, however the program does not mention to what protein atoms the mapping is done therefore, the authors prefer not to add this text to the graph as it would deliver more noise than relevant information.

30. (L. 387-394) It will be more beneficial to use auto docking programs and then analyze results, rather than manual docking and refinement (undescribed).

Thank you for this comment; we have tried default docking with MOE but the results were not fully to the author's satisfaction. All manipulations and steps are performed in the computational program MOE. Instead of using randomly generated poses or even constrained docking and scoring as done by the default docking procedure, the authors preferred the more user-controlled (manual) option, allowing to make sure that interactions with conserved residues are kept in the proposed fitting hypothesis and other interactions are being optimized by iterative constraint minimization procedures. We believe that this is the best way to create a realistic docking hypothesis, but we are also aware that there can be other binding hypotheses with such a flexible molecule. There is also no guarantee that the crystal structure, even after correcting for some transformations as described, has all the amino acids properly positioned to create the true binding pose.

31. (L.399) Are these residues conserved in SMOX and other polyamine oxidases?

In line 425 we address this point and a figure (Supplementary Figure 19) comparing hSMOX with other polyamine oxidases structures from different organism has been added, indicating the residues conserved.

32. (L. 407-412) SITE1 and SITE2 are mentioned but are not described in detail and not used for design of novel inhibitors. This paragraph is not informative and can be omitted.

We have added the label on supplementary figure 20 for further clarification and reference to supplementary figure 15. We consider the identification of a pocket specific to hSMOX very interesting for the further development of specific inhibitors.

33. (L. 420) See comment #11 above.

Ref

34. (L.424) "Asn313 ... occurs as Thr309": Substituted, replaced? See point #8 above.

The phrase has been reformulated for more clarity (line 215, 455)

35. (L.425) If the previous hypothesis about Glu196 and Ser198 is not supported by the structure, then how do these amino acids affect substrate specificity according to new structural data? Or the previous biochemical data are wrong/erroneous?

From the structure presented in this manuscript, it is not obvious to understand how Glu196 and Ser198 affect substrate specificity. It is also challenging to interpret previously published

results as wrong as we cannot exclude that residues mutated in the published work have a long-range effect on the substrate-binding pocket. We have better rephrased this concept in the manuscript (lines 456-461).

36. (L.429) How surface charge was calculated?

The surface charge is calculated with the default settings in MOE.

37. (L.440) "Spm was able to dock": Both Spm and JNJ-1289 can be docked simultaneously without steric clash?

Yes, the pockets do not overlap.

REVIEWERS' COMMENTS:

Reviewer #1 (Remarks to the Author):

The paper describes the first structure of SMOX along with hit finding and characterization of a hit, JNJ-1289. The structure of SMOX has been elusive with many groups failing to achieve it over the years. The protein engineering section presented here is very elegant and systematic. The hit finding campaign and subsequent characterization is also clear. The authors have addressed all comments from the initial review and should hence be publishable. However, I fail to see how the manuscript meets the criteria for publication in Comms bio. The hit finding and characterisation of the binding, including complex structure, makes for a good medicinal chemistry story, although the lack of cellular potency limits the value of JNJ-1289 as a starting point for further development or a tool compound. However, the general interest to the field of polyamine catabolism is limited. The reason for this is that, while the structure is the first one described for SMOX, it is somewhat artificial, with the MDL72527 bound in the structure. The more mechanistic insights into SMOX function is weak, building on biased docking of which is not underpinned by additional data. For me a better fit for this manuscript would be a medicinal chemistry journal, omitting the speculative section on spm binding and substrate selectivity.

Reviewer #2 (Remarks to the Author):

The authors have adequately addressed the reviewers' comments and the manuscript can be accepted for publication.

Reviewer #3 (Remarks to the Author):

The manuscript was significantly improved and all reviewer's concerns were fully addressed.